# Using the Classical Model for structured expert judgment to estimate extremes: a case study of discharges in the Meuse River

Guus Rongen[1,2], Oswaldo Morales-Nápoles[1], and Matthijs Kok[1,3]

[1]Civil Engineering and Geosciences, Delft University of Technology, The Netherlands
[2]Pattle Delamore Partners Ltd., New Zealand
[3]HKV consultants, The Netherlands

**Correspondence:** Guus Rongen (g.w.f.rongen@tudelft.nl)

**Abstract.**

Accurate estimation of extreme discharges in rivers, such as the Meuse, is crucial for effective flood risk assessment. However, hydrological models that estimate such discharges often lack transparency regarding the uncertainty of their predictions. This was evidenced by the devastating flood that occurred in July 2021 which was not captured by the existing model for esti-
mating design discharges. This article proposes an approach to obtain uncertainty estimates for extremes with structured expert judgment, using the Classical Model. A simple statistical model was developed for the river basin, consisting of correlated GEV distributions for discharges from upstream tributaries. The model was fitted to seven experts' estimates and historical measurements using Bayesian inference. Results fitted to only the measurements were solely informative for more frequent events, while fitting to only the expert estimates reduced uncertainty solely for extremes. Combining both historical observa-
tions and estimates of extremes provided the most plausible results. The Classical Model reduced the uncertainty by appointing most weight to the two most accurate experts, based on their estimates of less extreme discharges. The study demonstrates that with the presented Bayesian approach that combines historical data and expert-informed priors, a group of hydrological experts can provide plausible estimates for discharges, and potentially also other (hydrological) extremes, with a relatively manageable effort.

## 1   Introduction

Estimating the magnitude of extreme flood events comes with considerable uncertainty. This became clear once more on the 18th of July 2021: A flood wave on the Meuse River, following a few days of rain in the Eiffel and Ardennes, caused the highest peak discharge ever measured at Borgharen. Unprecedented rainfall volumes fell in a short period of time (Dewals et al., 2021). These caused flash floods with large loss of life and extensive damage in Germany, Belgium, and to a lesser extent also in the
Netherlands (TFFF, 2021; Mohr et al., 2022). The discharge at the Dutch border exceeded the flood events of 1926, 1993, and 1995. Contrary to those events, this flood occurred during summer, a season that is (or was) often considered less relevant for extreme discharges on the Meuse. A statistical analysis of annual maxima from a fact-finding study done recently after the flood, estimates the return period to be 120 years based on annual maxima, and 600 years when only summer half years (April to September) are considered (TFFF, 2021). These return periods were derived including the July 2021 event itself. Prior to

the event, it would have been assigned higher return periods. The season and rainfall intensity made the event unprecedented with regard to historical extremes. Given enough time, new extremes are inevitable, but with the Dutch flood safety standards being as high as once per 100,000 years (Ministry of Infrastructure and Environment, 2016) one would have hoped this type of event to be less surprising. The event underscores the importance of understanding the variability and uncertainty that comes with estimating extreme floods.

Extreme value analysis often involves estimating the magnitude of events that are greater than the largest from historical (representative) records. This requires establishing a model that described the probability of experiencing such events within a specific period, and subsequently extrapolating this to specific exceedance probabilities. For the Meuse, the traditional approach is fitting a probability distribution to periodic maxima and extrapolate from it (van de Langemheen and Berger, 2001). However, a statistical fit to observations is sensitive to the most extreme events in the time series available. Additionally, the hydrological

and hydraulic response to rainfall during extreme events might be different for more frequently occurring events, and therefore be incorrectly described by statistical extrapolation.

GRADE (Generator of Rainfall And Discharge Extremes) is a model-based answer to these shortcomings. It is used to determine design conditions for the rivers Meuse and Rhine in the Netherlands. GRADE is a variant on a conventional regional flood frequency analysis. Instead of using only historical observations, it resamples these into long synthetic time series of

rainfall that express the observed spatial and temporal variation. It then uses a hydrological model to calculate tributary flows and a hydraulic model to simulate river discharges (Leander et al., 2005; Hegnauer et al., 2014). Despite the fact that GRADE can create spatially coherent results and can simulate changes in the catchment or climate, it is still based on resampling available measurements or knowledge. Hence, it cannot simulate all types of events that are not present in the historical sample. This is illustrated by the fact that the July 2021 discharge was not exceeded once in the 50,000 years of summer discharges

generated by GRADE.

GRADE is an example where underestimation of uncertainty is observed, but certainly not the only model. For example, de Boer-Euser et al. (2017); Bouaziz et al. (2020) compared different hydrological modelling concepts for the Ourthe catchment (considered in this study as well) and showed the large differences that different models can give when comparing more characteristics than only stream flow. Regardless of the conceptual choices, all models have severe limitations when trying to

extrapolate to an event that has not occurred yet. We should be wary to disqualify a model in hindsight after a new extreme has occured. Alternatively, data-based approaches try to solve the shortcomings of a short record by extending the historical records with sources that can inform on past discharges. For example, paleoflood hydrology uses geomorphological marks in the landscape to estimate historical water levels (Benito and Thorndycraft, 2005). Another approach is to utilize qualitative historical written or depicted evidence to estimate past floods (Brázdil et al., 2012). The reliability of historical records can

be improved as well, for example by combining this with climatological information derived from more consistent sea level pressure data De Niel et al. (2017).

In this context, structured expert judgment (SEJ) is another data-based approach. Expert Judgment (EJ) is a broad term for gathering data from judgments based on expertise in a knowledge area or discipline. It is indispensable in every scientific application as a way of assessing the truth or value of new information. *Structured* expert judgment formalizes EJ by eliciting

expert judgments in such a way that judgments can be treated as scientific data. One structured method for this is the Classical Model, also known as *Cooke's method* (Cooke and Goossens, 2008). The Classical Model assigns a weight to each expert within a group (usually 5 to 10 experts) based on their performance in estimating the uncertainty in a number of seed questions. These weights are then applied to the experts' uncertainty estimates for the variables of interest, with the underlying assumption that the performance for the seed questions is representative for the performance in the questions of interest. Cooke and Goossens (2008) shows an overview of the different fields in which the Classical Model for structured expert judgment is applied. In total, data from 45 expert panels (involving in total 521 experts, 3688 variables, and 67,001 elicitations) are discussed, in applications ranging from nuclear, chemical and gas industry, water related, aerospace sector, occupational sector, health, banking, and volcanoes. Marti et al. (2021) used the same database of expert judgments and observed that using performance-based weighting gives more accurate DMs than assigning weights at random. Regarding geophysical applications, expert elicitation has recently been applied in different studies aimed at informing the uncertainty in climate model predictions (e.g., Oppenheimer et al., 2016; Bamber et al., 2019; Sebok et al., 2021). More closely related to this article, Kindermann et al. (2020) reproduced historical water levels using structured expert judgment (SEJ), and Rongen et al. (2022b) applied SEJ to estimate the probabilities of dike failure for the Dutch part of the Rhine River.

While examples of using specifically the Classical Model in hydrology are not abundantly available, there are many examples of expert judgment as prior information to decrease uncertainty and sensitivity. Four examples in which a Bayesian approach, similar to this study, was applied to limit the uncertainty in extreme discharge estimates are given by (Coles and Tawn, 1996; Parent and Bernier, 2003; Renard et al., 2006; Viglione et al., 2013). The mathematical approach varies between the different studies, but the rationale for using EJ is the same: adding uncertain prior information to the likelihood of available measurements to help achieve more plausible posterior estimates of extremes.

This study applies structured expert judgment to estimate the magnitude of discharge events for the Meuse River up to an annual exceedance probability of on average once per 1,000 years. We aim to get uncertainty estimates for these discharges. Their credibility is assessed by comparing them to GRADE, the aforementioned model-based method for deriving the Meuse River's design flood frequency statistics. A statistical model is quantified both with observed annual maxima and seven experts' estimates for the 10-year and 1000-year discharge on the main Meuse tributaries. The 10-year discharges (unknown to experts at the moment of the elicitation) are used to derive a performance-based expert weight that is used to inform the 1000-year discharges. Participants use their own approach to come up with uncertainty estimates. To investigate how the method that combines a) data and expert judgments compares to b) the data-only or c) the expert estimates-only approach, we quantify the model based on all three options. The differences show the added value of each component. This indicates the method's performance both when measurements are available and when they are not, for example in data scarce areas.

## 2 Study area and data used

Figure 1 shows an overview of the catchment of the Meuse River. The catchments that correspond to the main tributaries are outlined in red. The three locations for which we are interested in extreme discharge estimates, Borgharen, Roermond, and

Gennep, are colored blue. We call these 'downstream locations' throughout this study. The river continues further downstream until it flows into the North Sea near Rotterdam. This part of the river becomes increasingly intertwined with the Rhine River and more affected by the downstream sea water level. Consequently, the water levels can be ascribed decreasingly to the discharge from the upstream catchment. For this reason, we do not assess discharges further downstream than Gennep in this study.

The numbered dots indicate the locations along the tributaries where the discharges are measured. These locations' names and the tributaries' names are shown on the lower left. Elevation is shown with the grey-scale. Elevation data were obtained from EU-DEM (Copernicus Land Monitoring Service, 2017) and used to derive catchment delineation and tributary steepness. These data were provided to the experts together with other hydrological characteristics, like:

- *Catchment overview*: A map with elevation, catchments, tributaries, and gauging locations

- *Land use*: A map with land use from Copernicus Land Monitoring Service (2018)

- *River profiles and time of concentration*: A figure with longitudinal river profiles and a figure with time between the tributary peaks and the peak at Borgharen for discharges at Borgharen greater than 750 m$^3$/s.

- *Tabular catchment characteristics*, such as: Area per catchment, as well as the catchment's fraction of the total area upstream of the downstream locations. Soil composition from Food and Agriculture Organization of the United Nations (2003), specifying the fractions of sand, silt, and clay in the topsoil and subsoil. Land use fractions (paved, agriculture, forest & grassland, marshes, water bodies).

- *Statistics of precipitation*: Daily precipitation per month and catchment. Sum of annual precipitation per catchment. Intensity duration frequency curves for the annual recurrence intervals: 1, 2, 5, 10, 25, 50, and the maximum. All calculated from gridded E-OBS reanalysis data provided by Copernicus Land Monitoring Service (2020).

- *Hyetographs and hydrographs*: Temporal rainfall patterns and hydrographs for all catchments/tributaries during the 10 largest discharges measure at Borgharen (sources described below).

This information, included in the supplementary information, was provided to the experts to support them in making their estimates. The discharge data needed to fit the model to the observations were obtained from Service public de Wallonie (2022) for the Belgian gauges, Waterschap Limburg (2021); Rijkswaterstaat (2022) for the Dutch gauges, and Land NRW (2022) for the German gauge. These discharge data are mostly derived from measured water levels and rating curves. During floods, water level measurements can be incomplete and rating curves inaccurate. Consequently, discharge data during extremes can be unreliable. Measured discharge data were not provided to the experts, except in normalized form as hydrograph shapes.

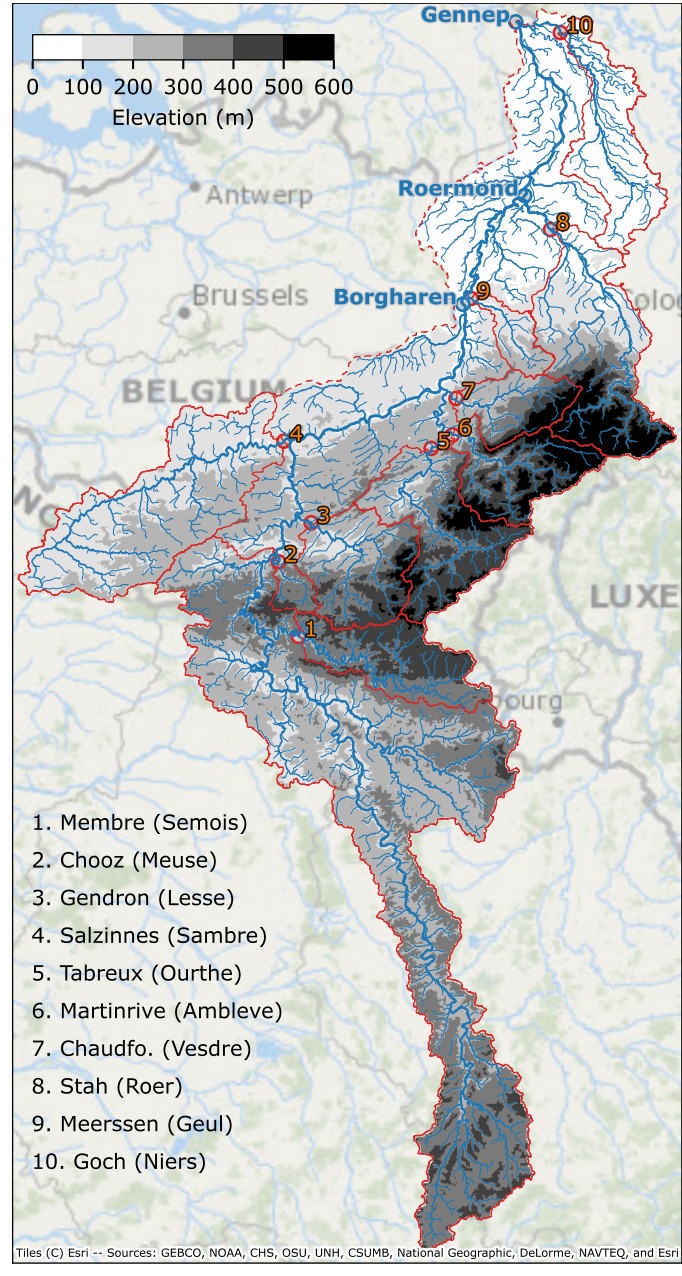

**Figure 1.** Map of the Meuse catchment considered in this study, with main river, tributaries, streams, and catchment bounds.

## 3 Method for estimating extreme discharges with experts

### 3.1 Probabilistic model

To obtain estimates for downstream discharge extremes, experts needed to quantify individual components in a model that gives the downstream discharge as the sum of the tributary discharges, times a factor correcting for covered area and hydrodynamics:

$$Q_d = f_{\Delta t} \cdot \sum_u Q_u, \tag{1}$$

where $Q_d$ is the peak discharge of a downstream location during an event, and $Q_u$ the peak discharge of the $u$'th (upstream) tributary during that event. Location $d$ can be any location along the river where the discharge is assumed to be dependent mainly on rainfall in the upstream catchment. The random variable $Q_u$ is modelled with the generalized extreme value (GEV)

distribution (Jenkinson, 1955). We chose this family of distributions firstly because it is widely used to estimate the probabilities of extreme events. Secondly, it provides flexibility to fit different rainfall-runoff responses by varying between Frechet (heavy tailed), Gumbel (exponential tail) and Weibull distributions (light tailed). We fitted the GEV distributions to observations, expert estimates, or both, using Bayesian inference (described in Sect. 3.3). The factor or ratio $f_{\Delta t}$ in Eq. 1 compensates for differences between the sum of upstream discharges and the downstream discharge. These result from, for example, hydraulic

properties such as the time difference between discharge peaks and peak attenuation as the flood wave travels through the river (which would individually lead to a factor $< 1.0$), or rainfall in the Meuse catchment area that is not covered by one of the tributaries (which would individually lead to a factor $> 1$). When combined, the factor can be lower or higher than $1$. The 1,000-year discharge is meant to inform the tail of the tributary discharge probability distributions. This tail is represented by the GEV tail shape parameter that is most difficult to estimate from data. We chose to elicit discharges, rather than a more

abstract parameter like the tail shape itself, such that experts make estimates on quantities that may be observed and at "a scale on which the expert has familiarity" (Coles and Tawn, 1996, p. 467).

The tributary peak discharges $Q_u$ are correlated because a rainfall event is likely to affect an area larger than a single tributary catchment and nearby catchments have similar hydrological characteristics. This dependence is modelled with a multivariate Gaussian copula that is realized through Bayesian Networks estimated by the experts (Hanea et al., 2015). The details of this

concern the practical and theoretical aspects of eliciting dependence with experts and are beyond the scope of this article. They will be presented in a separate article that is yet to be published. We did use the resulting correlation matrices for calculating the discharge statistics in this study. They are presented in appendix B.

In summary, using the method of SEJ described in Sect. 3.2, the experts estimate

1. the tributary peak discharges $Q_u$ that are exceeded on average once per 10 years and once per 1,000 years (for brevity

called the 10-year and 1,000-year discharge hereafter),

2. the factor $f_{\Delta t}$, and

3. the correlation between tributary peak discharges (as explained below).

With these, the model in Eq. 1 is quantified. The model was deliberately kept simple to ensure that the effect of the experts' estimates on the result remains traceable for them. Section 3.4 explains how downstream discharges were generated from these model components (i.e., the different terms in Eq. 1), including uncertainty bounds. The model is also described in more detail in (Rongen et al., 2022a) as well, where it was used in a data-driven context.

## 3.2 Assessing uncertainties using the Classical Model for expert judgments

The experts' estimates are elicited using the Classical Model. This is a structured approach to elicit uncertainty for unknown quantities. It combines expert judgments based on empirical control questions, with the aim to find a single combined estimate for the variables of interest (a rational consensus). The Classical Model is typically employed when alternative approaches for quantifying uncertain variables are lacking or unsatisfying (e.g., due to costs or ethical limitations). It is extensively described in (Cooke, 1991) while applications are discussed in (Cooke and Goossens, 2008). Here, we discuss the basic elements of the method. We applied the Classical Model because of its strong mathematical base, track record (Colson and Cooke, 2017), and the authors' familiarity with this method.

In the Classical Model, a group of participants, often researchers or practitioners in the field of interest, provides uncertainty estimates for a set of questions. These can be divided into two categories; seed and target questions. Seed questions are used to assess the participants' ability to estimate uncertainty within the context of the study. The answers to these questions are known by the researchers but not by the participants at the moment of the elicitation. Seed questions are often sourced from similar studies or cases and are as close as possible to the variables of interest. In any case, they are related to the field of expertise of the participant pool, but unknown to the participants. Target questions concern the variables of interest, for which the answer is unknown to both researchers as participants.

Because the goal is to elicit uncertainty, experts estimate percentiles rather than a single value. Typically, these are the 5th, 50th, and 95th percentile. Two scores are calculated from an expert's three-percentile estimates; the *statistical accuracy* (SA) and *information* score. The three percentiles create a probability vector with 4 inter-quantile intervals, $p = (0.05, 0.45, 0.45, 0.05)$. The fraction of realizations within each of expert $e$'s inter-quantile interval also forms a four-element vector $s(e)$. $s(e)$ and $p$ are expected to be more similar for an expert $e$ that correctly estimates uncertainty in the seed questions. The statistical accuracy is calculated by comparing each inter-quantile probability $p_i$ to $s_i(e)$. The SA is based on the relative information $I(s(e)|p)$, which equals $\sum_{i=1,...,4} s_i \log(s_i/p_i)$. Using the chi-square test (the quantity $2 \cdot N \cdot \sum_{i=1,...,4} s_i \log(s_i/p_i)$ is asymptotically $\chi_3^2$), the goodness-of-fit between the vectors $p$ and $s$ can be expressed as a p-value. This p-value is used as SA score. The SA is highest if the expert's probability-vector $s$ matches $p$. For twenty questions, this means the expert overestimates one seed question (i.e., the actual answer was below the 5th percentile), underestimates one question, and has nine questions in both the [5%, 50%] and [50%, 95%] interval. The further away the interquantile ratios $s_i/p_i$ are from 1.0, the lower the SA. Figure 4 is presented to visualize the disagreement between $s_i$ and $p_i$ for this study. This figure will be further discussed in subsection 4.1. For now, it is sufficient to note that the agreement between $s_i$ and $p_i$ is highest for expert D. The statistical accuracy expresses the ability of an expert to estimate uncertainty. Because a variable of interest is uncertain, its realization is considered to be a value sampled from the uncertainty distribution. According to the expert, this realization corresponds to a quantile on the

expert-estimated distribution. If an expert manages to reproduce the ratio of realizations within the interquantile intervals (such as in the example with 20 questions above), the probability of the expert being statistically accurate is high, hence they will receive a high p-value. Of course, this match could be coincidental, like any significant p-value from a statistical test. However, in general, a different sample of realizations (in this study, different observed 10-year discharges) is expected to give a p-value (i.e., statistical accuracy) of a similar order.

Additional to the SA, the information score compares the degree of uncertainty in an expert's answer compared to other experts. Percentile estimates that are close together (compared to the other participants) are more informative and get a higher information score. The product of the statistical accuracy and information score gives the expert's weight $w_\alpha(e)$:

$$w_\alpha(e) = 1_\alpha \times \text{statistical accuracy}(e) \times \text{information score}(e). \tag{2}$$

The statistical accuracy dominates the expert weight, where the information score modulates between experts with a similar SA. Experts with a SA lower than $\alpha$ can be excluded from the pool by using a threshold, expressed by the $1_\alpha$ in Eq. 2. This threshold is usually 5%. The (weighted) combination of the experts' estimates is called the decision maker (DM). The experts contribute to the $i$th item's DM estimate by their normalized weight:

$$\text{DM}_\alpha(i) = \sum_e w_\alpha(e) f_{e,i} \Big/ \sum_e w_\alpha(e). \tag{3}$$

This is called the global weight (GL) DM.

Alternatively, experts can be given the same weight which results in the equal weight (EQ) DM. This does not require eliciting seed variables, but neither does it distinguish experts based on their performance, a key aspect of the Classical Model (CM). Cooke et al. (2021) compare GL weights to EQ weights in an out-of-sample cross validation, and show that using performance-based weights increased the informativeness of the decision maker estimates by assigning weight to a few experts, without compromising the DM statistical accuracy (i.e., the performance of the DM in 'estimating' uncertainty).

To construct the DM, probability density functions (PDFs) such as $f_{e,i}$ in Eq. 3, need to be created from the percentile estimates. We used the Metalog distribution for this (Keelin, 2016). This distribution is capable of exactly fitting any three-percentile estimate. Notice that for this research, the Metalog distribution represents the uncertainty distribution of each expert over a particular discharge with a given return period. While it is related to the underlying distribution of discharge it does not make any assumption about this underlying distribution other than the ones expressed by experts through their percentile estimates. For symmetric estimates, the Metalog is bell-shaped. For asymmetric estimates, it becomes left- or right-skewed. Typically, the Classical Model assumes a uniform distribution in between the percentiles (minimum information). This leads to a stepped PDF where the Metalog gives a smooth PDF. An example of using the Metalog distribution in an expert elicitation study is described by Dion et al. (2020). All calculations related to the Classical Model were performed using the open-source software ANDURYL (Leontaris and Morales-Nápoles, 2018; 't Hart et al., 2019; Rongen et al., 2020).

In this study, the seed questions involve the 10-year discharges for the tributaries of the river Meuse. An example of a seed question is: "What is the discharge that is exceeded on average once per 10 years, for the Vesdre at Chaudfontaine?" The target questions concern the 1000-year discharges, as well as the ratio between the upstream sum and downstream discharge.

**Table 1.** List of experts with their affiliation and professional interests.

| Name | Affiliation | Field of expertise |
|------|-------------|--------------------|
| Alexander Bakker | Rijkswaterstaat & Delft University of Technology | Risk analysis for storm surge barriers, extreme value analyses, climate change and climate scenarios. |
| Eric Sprokkereef | Rijkswaterstaat | Coordinator crisis advisory group Rivers. Operational forecaster for Rhine and Meuse |
| Ferdinand Diermanse | Deltares | Expert advisor and researcher flood risk. |
| Helena Pavelková | Waterschap Limburg | Hydrologist |
| Jerom Aerts | Delft University of Technology | Hydrologist, focussed on hydrologic modelling on a global scale. PhD candidate. |
| Nicole Jungermann | HKV consultants | Advisor water and climate |
| Siebolt Folkertsma | Rijkswaterstaat | Advisor in the Team Expertise for the River Meuse |

Discharges with a 10-year recurrence interval are exceptional but can in general be reliably approximated from measured data. Seven experts participated in the in-person elicitation that took place on the 4th of July 2022. The study and model were discussed before the assessments to make sure that the concepts and questions were clear. After this, an exercise for the Weser catchment was done in which the experts answered four questions that were subsequently discussed. In this way, the experts could compare their answers to the realizations and view the resulting scores using the Classical Model.

Apart from the training exercise, the experts answered 26 questions: 10 seed questions regarding the 10-year discharge (one for each tributary), 10 target questions, regarding the 1,000-year discharge, and 6 target questions for the ratios between upstream sum and downstream discharge (10-year and 1,000-year, for three locations). A list of the seven participants' names, their affiliations, and their field of expertise is shown in Table 1. While the participants are pre-selected on their expertise, experts are scored *post hoc* in terms of their ability to estimate uncertainty in the context of the study. We note that the alphabetical order of the experts in the table does not correspond to their labels in the results. An overview of the data provided to the participants is given in Sect. 2, while the data itself, as well as the questionnaire, are presented in the supplementary information.

### 3.3 Determining model coefficients with Bayesian inference

The model for downstream discharges (Eq. 1) consists of generalized extreme value (GEV) distributions per tributary. The GEV-distribution has three parameters, the location ($\mu$), scale ($\sigma$), and shape parameter ($\xi$). Consider $z = (x - \mu)/\sigma$. The probability density function (PDF) of the GEV is then,

$$f(x) = \begin{cases} \frac{1}{\sigma} \exp\left(-\exp\left(-z\right)\right) \exp\left(-z\right), & \text{if } \xi = 0 \\ \frac{1}{\sigma} \exp\left(-(1 - \xi z)^{1/\xi}\right)(1 - \xi z)^{1/\xi - 1}, & \text{if } z \leq 1/\xi \text{ and } \xi > 0 \end{cases} \tag{4}$$

For each tributary, a (joint) distribution of the model parameters was determined using Bayesian inference, based on expert estimates and observed tributary discharge peaks during annual maxima at Borgharen. Bayesian methods explicitly incorporate uncertainty, a key aspect of this study, and provide a natural way to integrate expert judgment with observed data.

Bayes theorem gives the posterior distribution $p(\theta|\mathbf{q})$ of the (hypothesized) GEV-parameters $\theta$ given the observed peaks $\mathbf{q}$, as a function of the likelihood $p(\mathbf{q}|\theta)$ and the prior distribution $\pi(\theta)$:

$$p(\theta|\mathbf{q}) = \frac{p(\mathbf{q}|\theta)\pi(\theta)}{p(\mathbf{q})}. \tag{5}$$

The likelihood can be calculated using Eq. 4 from the product of the probability density of all (independent) annual maxima: $p(\mathbf{q}|\theta) = \prod_i \big(f(q_i|\theta)\big)$. The calculation of the prior is discussed below. That leaves $p(\mathbf{q})$, which is not straightforward to calculate. However, the posterior distribution can still be estimated using the Bayesian sampling technique Markov-Chain Monte Carlo (MCMC). MCMC algorithms compare different propositions of the numerator in Eq. 5, leaving the denominator as a normalization factor that crosses out. In this study, we used the affine invariant MCMC ensemble sampler as described by Goodman and Weare (2010), available through the Python module 'emcee' Foreman-Mackey et al. (2013). This sampler generates a trace of distribution parameters that forms the empirical joint probability distribution of, in our case, the three GEV parameters for each tributary. These are subsequently used to calculate the downstream discharges (see Sect. 3.4).

The prior consists of two parts, the expert estimates for the 10-year and 1,000-year discharge, and a prior for the GEV tail shape parameter $\xi$. Since the experts do not know the values of the discharges they are estimating, their estimates can be considered prior information. The prior probability $\pi(\theta)$ of the expert's estimates is calculated in a similar way as described by Viglione et al. (2013): Given a GEV-distribution $f(Q|\theta)$, the discharge $q$ for a specific annual exceedance probability $p$ is calculated from the quantile function or inverse CDF $(F^{-1})$,

$$q_{p_j} = F^{-1}(1 - p_j|\theta), \tag{6}$$

with $p_j$ being the $j$'th elicited exceedance probability. This discharge is compared to the expert's or DM's estimate for this 10- or 1,000-year discharge, $g(q_{p_j})$. Fig. 2 illustrates this procedure. The top curve $f(Q|\theta)$ represents a proposed GEV-distribution for the random variable $Q$ (tributary peak discharge) with parameter vector $\theta$. This GEV gives discharges corresponding to the 0.9 and 0.999th quantile (i.e., the 10-year and 1,000-year discharge). These discharges can then be compared to the expert estimates, illustrated by the two bottom graphs. Additionally, the figure shows the likelihood of observations with the vertical arrows ($p(\mathbf{q}|\theta)$ in Eq. 5).

Apart from the expert estimates, we prefer a weakly informative prior for $\theta$ (i.e., uninformative, but within bounds that ensure a stable simulation), such that only the data and expert estimates inform the final result. However, an informative prior was added to the shape parameter $\xi$ because with only expert estimates and no data, two discharge estimates are not sufficient for fitting the three parameters of the GEV-distribution. Additionally, the variance in the shape-parameter decreases with increasing number of years (or other block maxima) in a time series (Papalexiou and Koutsoyiannis, 2013). The 30 to 70 annual maxima per tributary in this study are not sufficient to reach convergence. Similar observations have been presented before for extreme precipitation in Koutsoyiannis (2004a, b) Therefore, we employ the geophysical prior as presented by Martins and Stedinger

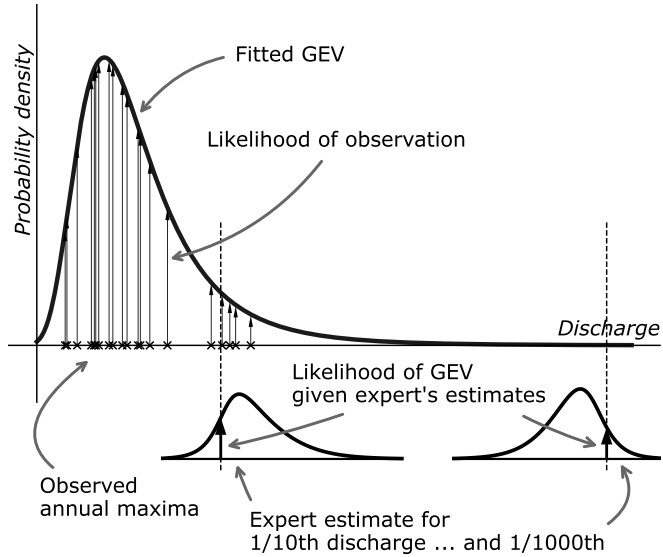

**Figure 2.** Conceptual visualization of elements in the likelihood-function of a tributary GEV-distribution.

(2000); a beta distribution with hyperparameters $\alpha = 6$ and $\beta = 9$ for $x \in [-0.5, 0.5]$, for which the PDF is:

$$h(x) = \frac{\Gamma(\alpha + \beta)}{\Gamma(\alpha)\Gamma(\beta)} x^{\alpha-1} (1-x)^{\beta-1}, \tag{7}$$

with $x = \xi + 0.5$, and $\Gamma$ being the gamma-function. This PDF is slightly skewed towards negative values of the shape parameter, preferring the heavy tailed Frechet distribution over the light tailed reversed Weibull. In their analysis of a very large number of

275 rainfall records worldwide, Papalexiou and Koutsoyiannis (2013) came to a similar distribution for the GEV-shape parameter. For $\mu$ and $\sigma$, we assigned equal probability to all values greater than 0. This corresponds to a weakly informative prior for $\mu$ (positive discharges), and an uninformative prior for $\sigma$ (only positive values are mathematically feasible).

With both expert estimates $g$ and the constrained tail shape, the prior distribution becomes

$$\pi(\theta) = \prod_j \left( g_j \left( F_\theta^{-1}(1-p_j) \right) \right) \cdot h(\xi + 0.5) \tag{8}$$

for $-0.5 < \xi < 0.5$, $\sigma > 0$, and $\mu > 0$. $\pi(\theta) = 0$ for any other combination. This gives all the components to calculate the posterior distribution in Eq. 5 using MCMC.

The posterior distribution comprises the prior tail-shape distribution, the prior expert estimates of the 10-year and 1,000-year discharges, and the likelihood of the observations. As described in Sect. 1 we compare the performance of using data, EJ, and the combination of both. If only data are used, the expert estimates drop out. If only expert judgments are used, the likelihood

drops out and both expert estimates are used. If both data and expert judgment are used, only the 1,000-year expert estimate is used.

With the just described procedure, the (posterior) distributions for the tributary discharges ($Q_u$ in Eq. 1) are quantified. This leaves the ratio between the upstream sum and downstream discharge ($f_{\Delta t}$) and the correlations between the tributary discharges to be estimated. For the ratios, we distinguished between observations and expert estimates as well. A log-normal distribution was fitted to the observations. This corresponds to a practical choice for a distribution of positive values with sufficient shape flexibility. The ratio itself does not represent streamflow, so there is no need to assume a heavy tailed distribution as would be expected for streamflow (Dimitriadis et al., 2021). The experts estimated a distribution for the factor as well, which was used directly for the experts-only fit. For the combined model fit, the observation-fitted log-normal distribution was used up to the 10-year range, and the expert estimate (fitted with a Metalog distribution) for the 1,000-year factor. Values of $f_{\Delta t}$ for return periods $T$ greater than 10 were interpolated (up to 1000-years) or extrapolated,

$$f_{\Delta t}|_T = f_{\Delta t}|_{10y} + \frac{\log(T) - \log(10)}{\log(1,000) - \log(10)} \cdot (f_{\Delta t}|_{1,000y} - f_{\Delta t}|_{10y}), \tag{9}$$

with $f_{\Delta t}|_{10y}$ being sampled from the lognormal and $f_{\Delta t}|_{1000y}$ from the expert estimated Metalog distribution. During the expert session, one participant requested to make different estimates for the factor at the 10-year event and 1,000-year event, a distinction that initially was not planned. Following this request, we changed the questionnaire such that a factor could be specified at both return periods. One expert used the option to make two different estimates for the factors.

Regarding the correlation matrix that describes the dependence between tributary extremes, the observed correlations were used for the data-only option and the expert-estimated correlations for the expert-only option. For the combined option, we took the average of the observed correlation matrix and the expert-estimated correlation matrix. Other possibilities for combining correlation matrices are available (see for example Al-Awadhi and Garthwaite, 1998, for a Bayesian approach), however in-depth research of these options is beyond the scope of this study.

### 3.4 Calculating the downstream discharges

The three components from Eq. 1 needed to calculate the downstream discharges are:

– Tributary (marginal) discharges, represented by the GEV-distributions from the Bayesian inference.

– The interdependence between tributaries, represented by a multivariate normal copula.

– The ratio between the upstream sum and downstream discharges ($f_{\Delta t}$).

In line with the objective of this article, an uncertainty estimate is derived for the downstream discharges. This section describes the method in a conceptual way. Appendix A contains a formal step-by-step description.

To calculate a single exceedance frequency curve for a downstream location, 10,000 events (annual discharge maxima) are drawn from the 9 tributaries' GEV-distributions. Note that 10 tributaries are displayed in Fig. 1. The Semois catchment is however part of the French Meuse catchment and therefore only used to assess expert performance. The 9 tributary peak discharges are summed per event and multiplied with 10,000 factors (one per event) for the ratio between upstream sum and downstream discharge. The 10,000 resulting downstream discharges are assigned an annual exceedance probability through

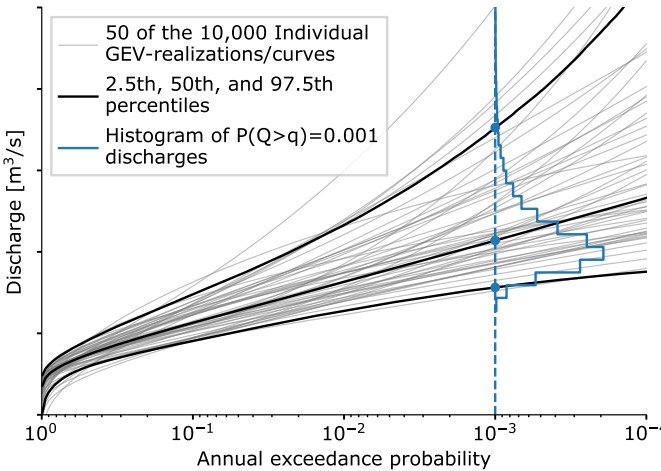

**Figure 3.** Individual exceedance frequency curves for each GEV-realization or downstream discharge, and the different percentiles derived from these.

empirical plot positions, resulting in an exceedance frequency curve. This process is repeated 10,000 times with different GEV-realizations from the MCMC-trace, resulting in 10,000 curves (each based on 10,000 discharges) from which the uncertainty bandwidth is determined. This is illustrated in Fig. 3. The grey lines depict 50 of the 10,000 curves (these can be both tributary GEV-curves, or downstream discharge curves). The (blue) histogram gives the distribution of the 1,000-year discharges. The colored dots indicate the 2.5th, 50th, and 97.5th percentiles in this histogram. Calculating these percentiles for all annual exceedance probabilities results in the black percentile curves, creating the uncertainty interval.

The dependence between tributaries is incorporated in two ways. First, the 10.000 events underlying each downstream discharge curve are correlated. This is achieved by drawing the $[9 \times 10,000]$ sample from the (multivariate normal) correlation model, transforming these samples to uniform space (with the normal CDF), and then to each tributary's GEV-distribution space (with the GEV's quantile function). This is the usual approach when working with a multivariate normal copula. The second way of incorporating the tributary dependence is by choosing GEV-combinations from the MCMC-results while considering the dependence between tributaries (i.e., picking high or low curves from the uncertainty bandwidth for multiple tributaries). As illustrated in Fig. 3, a tributary's GEV-distribution can lead to relatively low or high discharges. This uncertainty is largely caused by a lack of realizations in the tail (i.e., not having thousands of years of independent and identically distributed discharges). If one tributary would fit a GEV distribution resulting in a curve on the upper end of the bandwidth, it is likely because it experienced a high discharge event that affected its neighbouring tributary as well. Consequently, the neighbouring tributary is more likely to also have a 'high-discharge' GEV-combination. To account for this, we first sort the GEV-combinations based on their 1,000-year discharge (i.e., the curves' intersections with the blue dashed line), and draw a 9-sized sample from the dependence model. Transforming this to uniform space gives a value between 0 and 1 that is used as rank to select a (correlated) GEV-combination for each tributary. Doing this increases the likeliness that different tributaries will have relatively high or low sampled discharges.

**Table 2.** Scores for the Classical Model, for the experts (top 7 rows) and decision makers (bottom 3 rows).

| | Statistical accuracy | Information score | | Comb. score |
|---|---|---|---|---|
| | | All | Seed | |
| Exp A | 0.000799 | 1.605 | 1.533 | 0.00123 |
| Exp B | 0.000456 | 1.576 | 1.633 | 0.000745 |
| Exp C | $2.3 \cdot 10^{-8}$ | 1.900 | 1.868 | $4.4 \cdot 10^{-8}$ |
| Exp D | 0.683 | 0.711 | 0.626 | 0.427 |
| Exp E | 0.192 | 1.395 | 1.263 | 0.242 |
| Exp F | 0.000456 | 1.419 | 1.300 | 0.000593 |
| Exp G | 0.00629 | 1.302 | 1.232 | 0.00775 |
| GL (opt) | 0.683 | 0.659 | 0.670 | 0.458 |
| GL | 0.683 | 0.648 | 0.661 | 0.452 |
| EQ | 0.493 | 0.537 | 0.551 | 0.271 |

## 4   Experts' performance and resulting discharge statistics

This result section first presents the experts' scores for the Classical Model (Sect. 4.1) and the experts' rationale for answering the questions (Sect. 4.2). After this, the extreme value results for the tributaries (Sect. 4.3) and downstream locations (Sect. 4.4) are presented.

### 4.1   Results for the Classical Model

The experts estimated three-percentiles (5th, 50th and 95th) for the 10- and 1,000-year discharge for all larger tributaries in the

Meuse catchment. An overview of the answers is given in the supplementary material. Based on these estimates, the scores for the Classical Model are calculated as described in Sect. 3.2. The resulting statistical accuracy, information score, and combined score (which, after normalizing, become weights) are shown in table 2.

The statistical accuracy varies between $2.3 \cdot 10^{-8}$ for expert C to $0.683$ for expert D. Two experts have a score above a significance level of $0.05$. Figure 4 shows the position of each realization (answer) within the experts' three-percentile

estimate for each of the 10-year discharges. A high statistical accuracy means realizations to these seed variables are distributed accordingly to (or as close to) the mass in each inter-quantile bin: one realization below the 5th percentile, 4 in between the 5th and the median, four between the median and the 95th and one above the 95th. Expert D's estimates closely resemble this distribution ($\frac{1}{10}, \frac{5}{10}, \frac{4}{10}, \frac{0}{10}$ for each inter-quantile respectively), hence the high statistical accuracy score. A concentration of dots on both ends indicates overconfidence (too close together estimates, resulting in realizations outside of the 90% bounds).

We observe that most experts tend to underestimate the measured discharges, since most realizations are higher than their

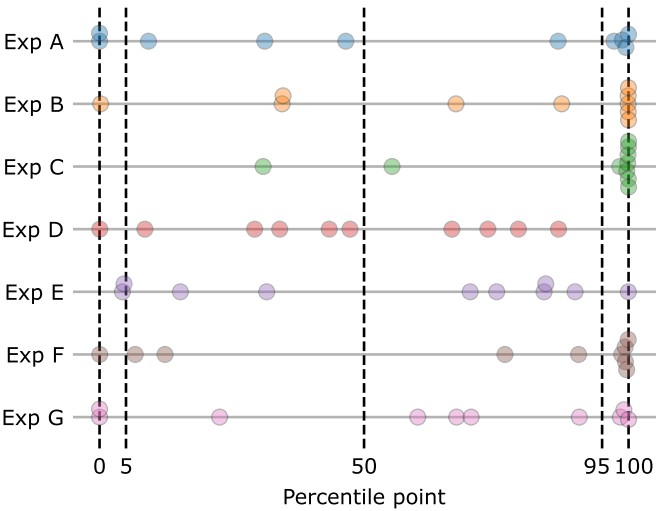

**Figure 4.** Seed question realizations compared to each expert's estimates. The position of each realization is displayed as percentile point in the expert's distribution estimate.

estimated 95th percentile. Note that the highest score is not received for the (median) estimates closest to the realization but to evenly distributed quantiles, as the goal is estimating uncertainty rather than estimating the observation (see Sect. 3.2).

The information scores show, as usual, less variation. The expert with the statistical accuracy (expert D) also has the lowest information score. Expert E, who has a high statistical accuracy as well, estimated more concentrated percentiles, resulting in a higher information score.

The variation between the three decision makers (DMs) in the table is limited. Optimizing the DM (i.e., excluding experts based on statistical accuracy to improve the DM-score) has a limited effect. In this case, only expert D and E would have a non-zero weight, resulting in more or less the same results compared to including all experts, even when some of them contribute with 'marginal' weights. The equal weights DM in this case results in an outcome that is comparable to that of the performance-based DM, i.e., a high statistical accuracy with a slightly lower information score compared to the other two DMs.

We present the model results as discussed earlier through three cases a) only data, b) only expert estimates, and c) the two combined as described in Section 3.3. We used the global weights DM for the data and experts option (c). This means the experts' estimates for the 10-year discharges were used to assess the value of the 1,000-year answer. For the experts-only option, we used the equal weights DM, because using the global weights emphasizes estimates matching the measured data in the 10-year range. This would indirectly lead to including the measured data in the fit. By using equal weights, we ignore the relevant seed questions and the corresponding differential weights.

## 4.2 Rationale for estimating tributary discharges

We requested the experts to briefly describe the procedure they followed for making their estimates. Overall, three approaches were distinguished. The first was using a simple conceptual hydrological model, in which the discharge follows from catchment characteristics like (a subset of) area, rainfall, evaporation and transpiration, rainfall-runoff response, land-use, subsoil, slope, or the presence of reservoirs. Most of this information was provided to the experts, and if not, they made estimates for it themselves. A second approach was to compare the catchments to other catchments known by the expert, and possibly adjusting the outcomes based on specific differences. A third approach was using rules of thumb, such as the expected discharge per square kilometer of catchment or a 'known' factor between an upstream tributary discharge and a downstream discharge (of which the statistics are better known). For estimating the 1,000-year discharge, the experts had to do some kind of extrapolation. Some experts scaled with a fixed factor, while others tried to extrapolate the rainfall, for which empirical statistics where provided. The hydrological data (described in Sect. 2) was provided to the experts in spreadsheets as well, making it easier for them to do computations. However, the time frame of one day (for the full elicitation) limited the possibilities for making detailed model simulations.

Figure 5 shows how the different approaches led to different answers per tributary. It compares the 50th percentile of the discharge estimates per tributary of each expert, by dividing them through the catchment area. The 10-year and 1,000-year discharges from fitting the observations (i.e., the data only approach) are indicated with the starts. The figure shows that most experts estimated higher discharges for the steeper tributaries (Ambleve, Vesdre, Lesse). The experts estimated the median 1,000-year discharges to be 1.7 to 3.8 times as high as the median 10-year discharge, with an average of on average 2.3 for all experts and tributaries. The statistically most accurate expert, Expert D, estimated factors in between 1.6 and 7.0. Contrarily, expert E, with the second highest score, estimated a ratio of 2.0 for all tributaries. For estimating the factor between the tributaries' sum and the downstream discharge ($f_{\Delta t}$ in Eq. 1), experts mainly took into consideration that not 100% of the area is covered by the tributary catchments for which the discharge-estimates were made, and that the tributary hydrograph peaks have different lag times. Additional aspects noted by the experts were the effects of flood peak attenuation and spatial dependence between tributaries and rainfall.

## 4.3 Extreme discharges for tributaries

We calculated the extreme discharge statistics for each of the tributaries based on the procedures described in Sect. 3.3. Figure 6 shows the results for Chooz and Chaudfontaine (left and middle column). Chooz is a larger not too steep tributary, while Chaudfontaine is a smaller steep tributary (see figure 1). The right column shows the discharges for Borgharen, the location where we want to estimate the discharges through Eq. 1, which is further discussed in Sect. 4.4. The results for the other tributaries are shown in the supplementary information for all experts and DMs.

The top row (a, d, g) in Fig. 6 shows the uncertainty interval of these distributions when fitted only to the discharge measurements. The outer colored area is the 95% interval, the opaquer inner area the 50% interval, and the thick line the median value. The second row (b, e, h) shows the fitted distributions when only expert estimates are used. The bottom row (c, f, i) shows

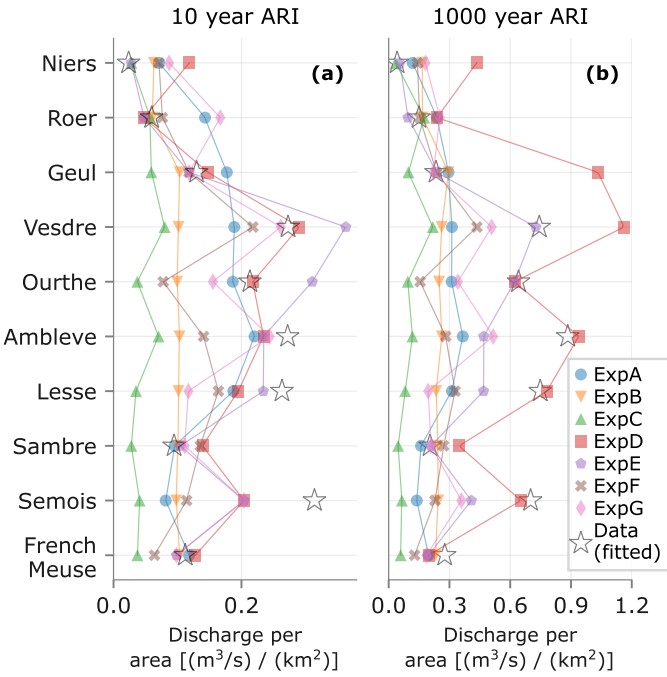

**Figure 5.** Discharge per area for each tributary and experts, based on the estimate for the 50th percentile. **(a)** for the 10-year, and **(b)** for the 1,000-year discharge. Observed or fitted discharges are indicated with stars. The lines are displayed to help distinguish overlapping markers.

the combination of expert estimates and data. The data-only option closely matches the data in the return period range where data are available, but the uncertainty interval grows for return periods further outside sample. Contrarily, the experts-only option shows much more variation in the 'in sample' range, while the out of sample return periods are more constrained. The combined option is accurate in the 'in sample' range, while the influence of the DM estimates is visible in the 1,000-year return period range.

## 4.4 Extreme discharges for Borgharen

Combining all the marginal (tributary) statistics with the factor for downstream discharges and the correlation models estimated by the experts, we get the discharge statistics for Borgharen. The results for this are shown in Fig. 6 (g, h, i).

As with the statistics of the tributaries, we observe high accuracy for the data-only estimates in the 'in sample' range, constrained uncertainty bounds for EJ-only in the range with higher return periods, and both when combined. The combined results match the historical observations well. Note that this is not self-evident as the distributions were not fitted directly to the observed discharges at Borgharen but rather obtained through the dependence model for individual catchments and equation 1. Contrarily, the data-only results deviate from the observations in the 10- to 100-year range. Sampling from the fitted model components (GEVs, dependence model, and factors) does not accurately reproduce the downstream discharges in this range because they were individually fitted and not as a whole. We do not consider this a problem, as the study is oriented

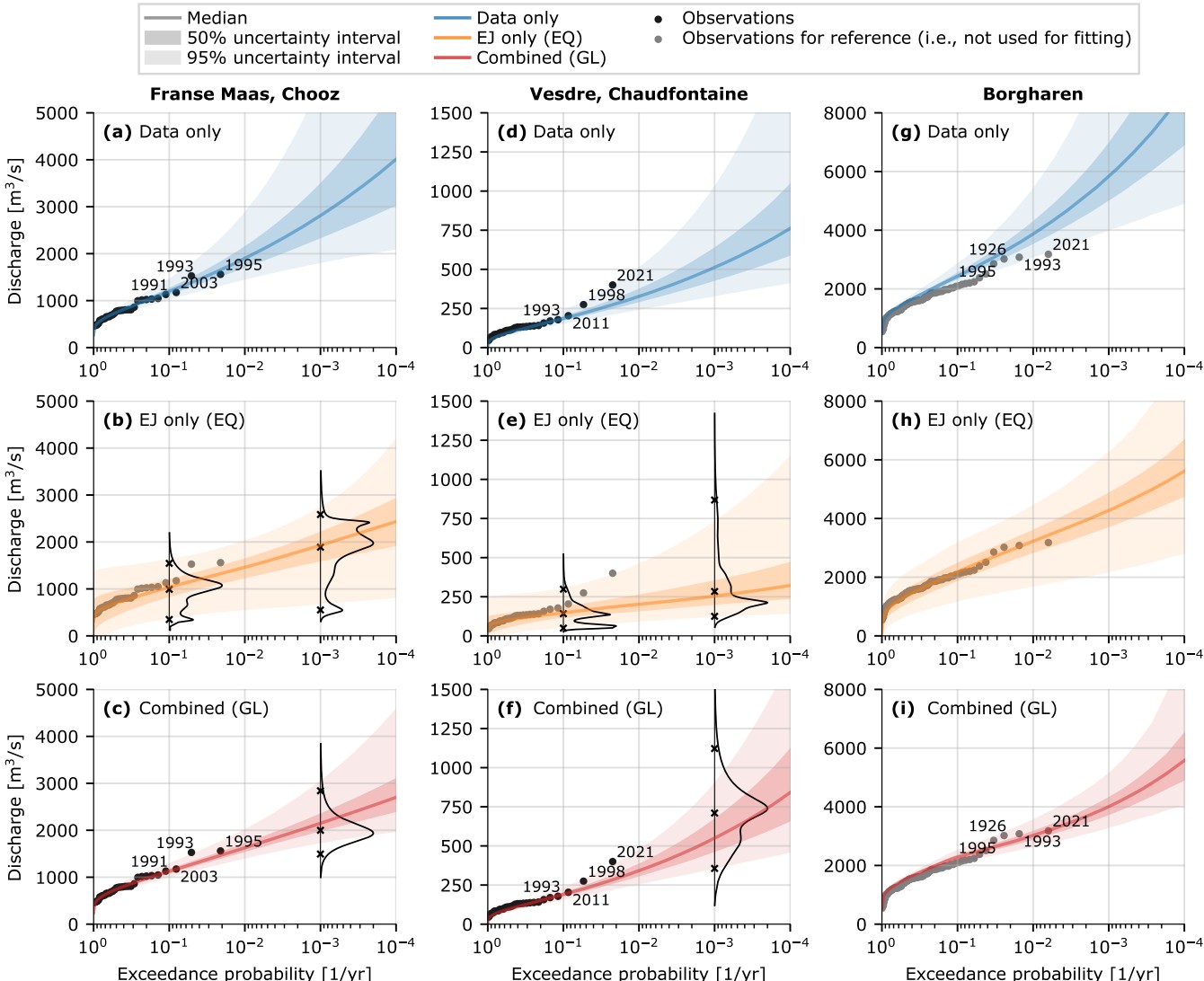

**Figure 6.** Extreme discharge statistics for Chooz (**a, b, c**), Chaudfontaine (**d, e, f**) and Borgharen (**g, h, i**). (**a, d, g**) represent data only, (**b, e, h**) expert judgment only, and (**c, f, i**) the data and expert judgment combined.

towards showing the effects of expert quantification in combination with more traditional hydrological modelling. The EJ-only estimates give a much wider uncertainty estimate. The experts' combined median matches the observations surprisingly well, but the large uncertainty within the observed range cautions against drawing general conclusions on this.

Zooming in on the discharge statistics for the downstream location Borgharen, we consider the 10, 100, and 1,000-year dis-
charge. Figure 7 shows the (conditional) probability distributions (smoothed with a kernel density estimate) for these discharges at the location of interest.

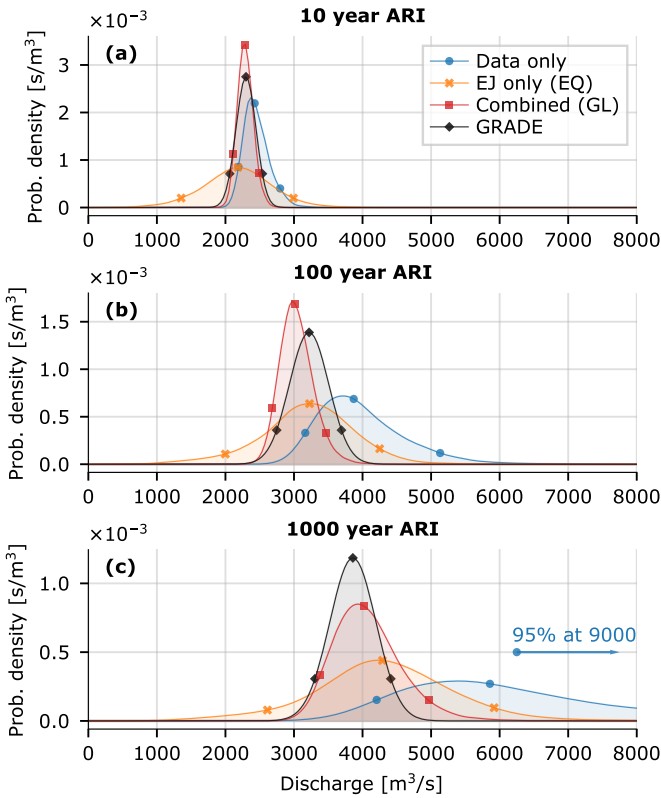

**Figure 7.** Kernel density estimates for the 10-year **(a)**, 100-year **(b)**, and 1,000-year **(c)** discharge for Borgharen. The dots indicate the 5th, 50th and 95th percentile.

Comparing the three modelling options discussed thus far, we see that the data-only option is very uncertain, with a 95% uncertainty interval of 4,000 to around 9,000 m$^3$/s for the 1,000-year discharge. A Meuse-discharge of 4,000 m$^3$/s will likely flood large stretches along the Meuse in the Dutch province Limburg, while a discharge of 5,000 m$^3$/s also floods large areas further downstream (Rongen, 2016). For discharges higher than 6,000 m$^3$/s the applied model (Eq. 1) should be reconsidered, as the hydrodynamic properties of the system change due to upstream flooding.

The combined results are surprisingly close to the currently used GRADE-statistics for dike assessment; the uncertainty is slightly larger, but the median is very similar. The EJ-only results are less precise, but the median values are similar to the combined results and GRADE-statistics. The large uncertainty is mainly the results of equally weighting all experts, instead of assigning most weight to experts D and E (as done for the global weight DM). For the combined data and EJ approach, the results for the tributary discharges roughly cover the intersection of the EJ-only and data-only results (see Fig. 6 a-f). Figure 7 does not show this pattern, with the EJ-only results positioned in between the data-only and combined results. This is mainly due to equal weight DM used for the EJ-only results, which gives a higher factor between upstream and downstream discharges ($f_{\Delta t}$ in Eq. 1), and therefore higher resulting downstream discharges. Overall, the combined effect of data and EJ

is more difficult to identify in the downstream discharges (Fig. 6 g-i) than it is in the tributary discharge GEVs (Fig. 6 a-f). This is due to the additional model components (i.e., the factor between upstream and downstream, and the correlation model) affecting the results. Additional plots similar to Fig. 6 that illustrate this are presented in the supplementary information. There, the results for the other two downstream locations, Roermond and Gennep, are presented as well. These results behave similar to those for Borgharen and are therefore not presented here.

## 5    Discussion


This study proposed a method to estimate credible discharge extremes for the Meuse River (1,000-year discharges in the case of this research). Observed discharges were combined with expert estimates through the GEV-distribution, using Bayesian inference. The GEV-distribution has typically less predictive power in the extrapolated range. Including expert estimates, weighted by their ability to estimate the 10-year discharges, improved the precision in this range of extremes.

Several model choices were made to obtain these results. Their implications warrant further discussion and substantiation. This section addresses the choice for the elicited variables, the predictive power of 10-year discharge estimates for 1000-year discharges, the overall credibility of the results, and finally, some comments on model choices and uncertainty.

### 5.1    Method and model choices

We chose to elicit tributary discharges, rather than the downstream discharges (our ultimate variable of interest) themselves.
We believe that experts' estimates for tributary discharges correspond better to catchment hydrology (rainfall-runoff response). Additionally, this choice enables us to validate the final result with the downstream discharges. With the chosen set-up we thus test the experts' capabilities for estimating system discharge extremes from tributary components, while still considering the catchment hydrology, rather than just informing us with their estimates for the end results. However, this does not guarantee that the downstream discharges calculated from the experts' answers match the discharges they would have given if elicited
directly.

     We fitted the GEV-distribution based on the elicited 10-year and 1000-year discharges. In particular the GEV's uncertain tail shape parameter is informed through this, as the location and scale parameter can be estimated from data with relative certainty. Alternatively, we could have estimated the tail shape parameter directly or estimated a related parameter such as the ratio or difference between discharges. The latter was done by Renard et al. (2006) who elicited the 10-year discharge and the
*differences* between the 10- and 100-year and 100- and 1,000-year discharges. This approach reduces the dependence between expert estimates for different quantiles, and therefore between the priors (when more than one quantile is used) (Coles and Tawn, 1996). Additionally, it shifts the experts' focus to assessing how surprising or extreme rare events can be. Because we were ultimately interested in the 1000-year discharges, we chose eliciting this discharge directly. This will give a more accurate representation of this specific value than composing it of two random variables with a dependence that is unknown to us. We
appreciate however that if experts would have estimates ratios or differences, and been evaluated by this, different weights would have resulted than the ones presented in this research (refer to the markedly different ratios between the 10-year and

1,000-year discharge for the two best experts D and E in Fig. 5). A study focusing on how surprising large events can be, and whether one method renders consistently larger estimates than the other, would make an interesting comparison. Finally, we note that Renard et al. (2006) combine different extreme value distributions with non-stationary parameters in a single Bayesian analysis, which makes their method a good example of incorporate climate change effects (often considered a driver of for new extremes) in the method as well. This was however out of the scope of our research, which shows that extreme discharge statistics can be improved when combining them with structured expert judgment procedures.

Regarding the goodness-of-fit of the chosen GEV distribution, we note that some of the experts estimated 1,000-year discharges much higher of lower than would be expected from observations. This might indicate that the GEV-distribution is not the right model to observations and expert estimates. However, a significantly lower estimate indicates that the estimated discharge is wrong, as it is unlikely that the 1,000-year discharge is lower than the highest on record. A significantly higher estimate, on the other hand, might be valid, due to a belief in a change in catchment response under extreme rainfall (e.g., due to a failing dam). This would violate the GEV-distribution's 'identically distributed' assumption. However, the GEV has sufficient shape flexibility to facilitate substantially higher 1,000-year discharges, so we do not consider this a realistic shortcoming. Accordingly, rather than viewing the GEV as a limiting factor for fitting the data, we use it as a validation for the Classical Model scores, as described in Sect. 5.2.

Finally, we note the model's omission of seasonality. The July 2021 event was mainly extraordinary because of its magnitude *in combination with* the fact that it happened during summer. Including seasonality would have been a valuable addition to the model but it would also have (at least) doubled the number of estimates provided by each expert, which was not feasible for this study. The exclusion of seasonality from our research does not alter our main conclusion, which is the possibility of enhancing estimation of extreme discharges through structured expert judgments.

## 5.2 Validity of the results

The experts participating in this study were asked to estimate 10-year and 1000-year discharges. While both discharges are unknown to the expert, the underlying processes leading to the different return period estimates can be different. An implicit assumption is that the experts' ability to estimate the seed variables (a 10-year discharge) reflects their ability to estimate the target variables (a 1000-year discharge). This assumption is in fact one of the most crucial assumptions in the Classical Model. The objective of this research is not to investigate this assumption. For an example of a recent discussion on the effect of seed variables on the performance of the Classical Model the reader is referred to Eggstaff et al. (2014). The representativeness of the seed variables for calibration variables has extensively been discussed in, for example, Cooke (1991). Seed questions have to be as close as possible to the variables of interest, and mostly concern similar questions from different cases or studies. Precise 1000-year discharge estimates are however unknown for any river system, making this option infeasible for this study. In comparison, with a conventional model-based approach, the ability of a model to predict extremes is also estimated from (and tailored to) the ability to estimate historical observations (through calibration). Advantages of relying in the extrapolation of a group of experts are that they can explicitly consider uncertainty and are assessed on their ability to do so through the Classical Model. In Sect. 5.1 we described how inconsistencies between the observations and expert estimates can lead to a

sub-optimal GEV-fit. The fact that this is most prevalent in the low-scoring experts and least for experts D and E supports the credibility of the results. Moreover, this means that the 'bad' fits have little weight in the final global weight DM results, and secondly that the GEV is considered a suitable statistical distribution to fit observations and expert estimates.

The GRADE results from (Hegnauer and Van den Boogaard, 2016) were used to validate the 1,000-year downstream discharge results. These GRADE-statistics at Borgharen (currently used for dike assessment) give a lower and less uncertain range for the 1,000-year discharge than the estimates obtained through our methodology. The estimates obtained in this study present larger uncertainty bands and indicate higher extreme discharges. This might be a consequence of the fact that we did not show the measured tributary discharges to the experts, such that we could clearly distinguish the effect of observations and 'prior' expert judgments. Moreover, GRADE (at the time) did not include the July 2021 event. If the GRADE statistics had been derived with the inclusion of the July 2021 event, it would likely assign more probability to higher discharges. The experts' estimates on the contrary were elicited after the July 2021 event which likely did affect their estimates. Therefore, the comparison between GRADE and the expert estimates should not be used to assess correctness, but as an indication of whether the results are in the right range. Finally, note that the full GRADE-method is not published in a peer-reviewed journal (the weather generator is, (Leander et al., 2005)). However, because the results are widely used in the Dutch practice of flood risk assessment (and known to the experts as well) we considered them the best source for comparing the results in the present study.

To evaluate the value of the applied approach that uses data combined with expert estimates, we compared the results that were fitted to only data or only expert judgment to the results of the combination. For the last option we used an equal weight decision maker, a conservative choice as the experts' statistical accuracy could potentially still be determined based on a different river where data for seed questions are available. While the marginal distributions of the EJ-only case present wide bandwidths (see Fig. 6 b and e), the final results for Borgharen still gave a statistically accurate result but with a few caveats, namely that the uncertainty is very large and that the 10-year and 1,000-year estimates in itself are insufficient to inform the GEV without adding prior information (otherwise we have 2 estimates for 3 parameters). Consequently, when only using expert estimates, eliciting the random variable (discharges) directly through a number of quantiles of interest, might be a suitable alternative.

### 5.3 Final remarks on model choices

Finally, we note that using expert judgment to estimates discharges through a model (like we did) still gives the analyst a large influence in the results. We try to keep the model transparent and provide the experts with unbiased information, but by defining the model on beforehand and providing specific information we steer the participants towards a specific way of reasoning. Every step in the method, such as the choice for a GEV-distribution, the dependence model, or the choice for the Classical Model, affects the end result. By presenting the method and providing background information explicitly, we hope to have made this transparent and show the usefulness of the method for similar applications.

## 6   Conclusions

This study sets out to establish a method for estimation of statistical extremes through structured expert judgment and Bayesian inference, in a case-study for extreme river discharges on the Meuse River. Experts' estimates of tributary discharges that are exceeded in a once per 10 year and once per 1,000-year event are combined with high river discharges measured over the past 30-70 years. We combine the discharges from different tributaries with a multivariate correlation model describing their dependence and compare the results for three approaches, a) data only, b) expert judgment only, and c) the combination. The expert elicitation is formalized with the Classical Model for structured expert judgment.

The results of applying our method show credible extreme river discharges resulting from the combined expert-and-data approach. A comparison to GRADE, the prevailing method for estimating discharge extremes on the Meuse, gives similar ranges for the 10-, 100-, 1,000-year discharges as GRADE. Moreover, the two experts with the highest scores from the Classical Model had discharge estimates that correspond well with those discharges that might be expected from the observations. This indicates that using the Classical Model to assess expert performance is a suitable way of using expert judgment to limit the uncertainty in the "out of sample" range of extremes. The experts-only approach performs satisfactory as well, albeit with a considerably larger uncertainty than the EJ-data option. The method may also be applied to river systems where measurement data are scarce or absent, but adding information on less extreme events is desirable to increase the precision of the estimates.

On a broader level, this study has demonstrated the potential of combining structured expert judgment and Bayesian analysis in informing priors and reducing uncertainty in statistical models. When estimates on uncertain extremes are needed, which cannot satisfactorily be derived (exclusively) from a (limited) data-record, the presented approach provides a means (not the only mean) of supplementing this information. Structured expert judgment provides an approach of deriving defensible priors, while the Bayesian framework offers flexibility for incorporating these into probabilistic results by adjusting the likelihood of input or output parameters. In our application to the Meuse River, we successfully elicited credible extreme discharges. However, case studies for different rivers should verify these findings. Our research does not discourages the use of more traditional approaches such as rainfall-runoff or other hydrodynamic or statistical models. Considering the credible results and the relatively manageable effort required, the approach (when well implemented) can present an attractive alternative to models that approach uncertainty in extremes in a less transparent way.

## Appendix A:   Calculation of downstream discharges

Section 3.4 explained the method applied and choices made for calculating downstream discharges. This appendix explains this in more detail, including the mathematical equations.

Three model components are elicited from the experts and data:

- Marginal tributary discharges, in the form of a MCMC GEV-parameter trace. Each combination $\theta$ consists of a location ($\mu$), scale ($\sigma$), and tail-shape parameter ($\xi$).

– A ratio between the sum of upstream peak discharges and the downstream peak discharge, represented by This is a single probability distribution.

– The interdependence between tributary discharges, in the form of a multivariate normal distribution.

The exceedance frequency curves for the downstream discharges are calculated based on 9 tributaries ($N_T$), a trace of 10,000 MCMC parameter combinations ($N_M$), and 10,000 discharge events ($N_Q$) per curve.

The $N_M$ parameter combinations for each tributary are sorted based on the (1,000-year) discharge with an exceedance probability of 0.001: $F_{GEV}^{-1}(1 - 0.001|\theta)$, in which $F_{GEV}^{-1}$ is the inverse cumulative density function, or percentile point function, of the tributary GEV. Sorting the discharges like this enables us to select parameter combinations that lead to low or high discharges in multiple tributaries, and in this way express the tributary correlations. The sorting order might be different for the 10-year discharge than it is for the 1000-year discharge. The latter is however chosen as it is most interesting for this study.

For calculating a single curve, $N_T$ realizations are drawn from the dependence model. These normally distributed realizations ($\mathbf{x}$) are transformed to the $[1, N_M]$ interval, and are then used as index $\mathbf{j}$ to select a GEV-parameter combination for each of the $N_T$ tributaries:

$$\mathbf{j} = Round(F_{norm}(\mathbf{x}) \cdot (N_M - 1) + 1). \tag{A1}$$

This is the first of two ways in which the interdependence between tributary discharges is expressed. The second is the next step, drawing a ($N_T \times N_Q$) sample $\mathbf{Y}$ from the dependence model. These events (on a standard normal scale) are transformed to the discharge realizations $\mathbf{Q}$ for each tributary's GEV parameter combination:

$$\mathbf{Q} = F_{GEV,\mathbf{j}}^{-1}(F_{norm}(\mathbf{Y})) \tag{A2}$$

An $N_Q$ sized sample for the ratio between upstream sum and downstream discharges ($\mathbf{f}$) is drawn as well. The ($N_T \times N_Q$) discharges $\mathbf{Q}$ are summed per event (for all tributaries), and multiplied with the factor $\mathbf{f}$,

$$\mathbf{q} = \mathbf{f} \cdot \sum(\mathbf{Q}). \tag{A3}$$

Note that this notation corresponds to Eq. 1. The $N_Q$ discharges $\mathbf{q}$ are subsequently sorted and assigned a plot positions:

$$\mathbf{p} = \frac{\mathbf{k} - a}{N_Q + b}, \tag{A4}$$

with $a$ and $b$ being the plot positions, 0.3 and 0.4, respectively (from Bernard and Bos-Levenbach, 1955). $\mathbf{k}$ indicates the order of the events in the set (1 being the largest, $N_Q$ the smallest), The plot positions ($\mathbf{p}$) are the 'empirical' exceedance probabilities of the model. With 10,000 discharges and our exceedance probability of interest of 1/1,000, the results are insensitive to the choice of plot positions.

This procedure results in one exceedance frequency curve for the downstream discharge. The procedure is repeated 10,000 times to generate an uncertainty interval for the discharge estimate. Note that the full Monte Carlo simulation comprises $10,000 \times 10,000 = 100,000,000$ 'events' for the 9 tributaries.

## Appendix B: Expert and DM correlation matrices

Figure B1 shows the correlation matrices estimated by the experts. The DM correlation matrices are weighted combinations of the expert matrices, based on the weights from Table 2. See subsection 3.2 and equation 3.

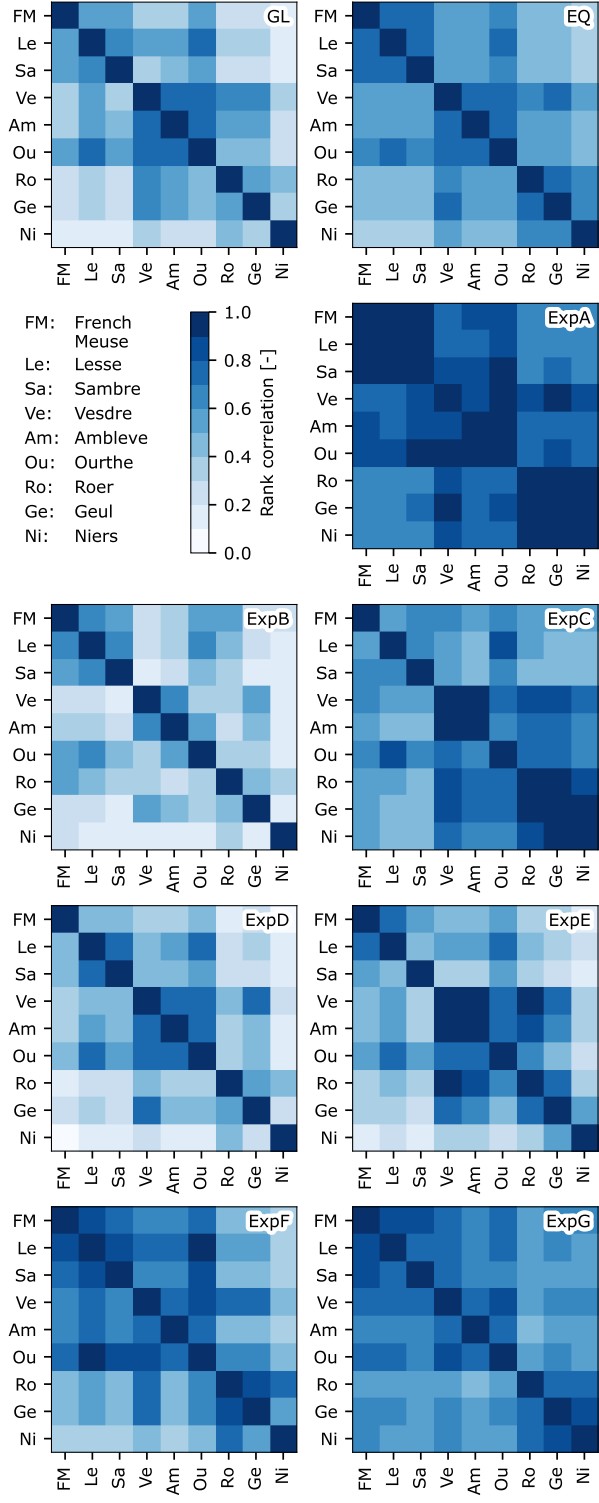

**Figure B1.** Correlation matrices estimated by the expert

*Code and data availability.* The data and the code used to process are openly available under the GNU license through: https://github.com/ grongen/meuse_extremes_sej

*Author contributions.* GR, OMN, and MK planned the study. GR prepared and carried out the expert elicitation, processed and analyzed the
results, and wrote the manuscript draft. OMN and MK reviewed and edited the manuscript.

*Competing interests.* The authors declare that they have no conflict of interest.

*Acknowledgements.* We would like to thank all experts that participated in the study, Alexander, Eric, Ferdinand, Helena, Jerom, Nicole, and Siebolt, for their time and effort in making this research possible. Secondly, we thank Dorien Lugt en Ties van der Heijden, who's hydrological and statistical expertise greatly helped in preparing the study through test rounds.
This research was funded by the TKI project EMU-FD. This research project is funded by Rijkswaterstaat, Deltares and HKV consultants.

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
