# Peer review of "Using the Classical Model for structured expert judgment to estimate extremes: a case study of discharges in the Meuse River"

_EGUsphere, 2023_

## Author Comment (AC1)

First of all, the authors would like to thank the reviewer for the detailed and constructive review. The reviewer mentions valid points. We (the authors) have added a response to each of them. If the reviewer agrees with our interpretation of the comments and the responses, we will solve most be making textual changes. The two main points that deviate from this are:

- Regarding the prior, we suggest doing more than just textual changes: We suggest changing the prior weakly informed prior described in Appendix A for an uninformed prior on location and scale, and an informed prior on the shape parameter. A small test shows this give similar good results, while it simplifies the method regarding the prior.
- While the suggestion of using the expert judgments as prior makes sense from a Bayesian statistics point of view, it would reduce the effect of expert judgments in this EJ-study. Therefore we choose not to do so. Hopefully this will be clarified after reading the full response.

The paper provides the result of an interesting experiment in which flood experts are asked to guess the flood frequency curves for several sites in a region without access to discharge data, and the information is then used together with observed maximum annual flood peaks to improve the "credibility" of the estimated tails of the distributions. A procedure is developed to use expert opinions on several tributaries and transform them into an estimate for a downstream gauge.

The paper is original, as far as I can tell, and deals with an important issue in flood risk assessment, i.e., the formal use of expert opinion in flood frequency analysis. Even though I liked reading the paper, there are some parts that, in my opinion, need to be improved, clarified, better explained or discussed before publication. My main concerns are:

1) I am not sure that the ability of the expert in providing a judgement on the flood frequency curve can be measured by her/his ability in guessing the 10-yr flood in absolute terms. If one wants the expert to help in reducing uncertainty in the tails of the distribution, she/he should inform us on how large floods may compare to small floods, by reasoning on the driving processes. In the end, it is the shape of the flood frequency distribution that's hard to get with local data, not the location. The proposed method seems to be tailored for getting the order of magnitude right, i.e., the flood magnitude in m^3/s, but not how surprising can large extreme events be compared to the more frequent ones.

This is a valid point, and a fundamental point to SEJ in general. We are trying to assess the expert's ability of estimating out of sample events from their ability to estimate events in the frequent/observed range. There are different approaches possible for doing this. Our choice is in line with Cooke's philosophy, that says expert judgments should be as little perturbed as possible. This means using answers close to how they are estimating, which means it is preferable to estimate the 1000-year event directly than estimating a ratio or tail shape and deriving the 1000-year discharge (or other extremes) from that. This is further discussed in the detailed comments below.

2) the expert information is accounted for as data (part of the likelihood) using an ad-hoc procedure, which seems to me inconsistent with the Bayesian way. Why not accounting for expert judgement as prior information? That would be the natural Bayesian way to do it: since the experts give their estimates without using discharge data, this can be considered as prior information.

I agree that considering the expert judgments as priors, and the observations as data, would be a more Bayesian approach. It does however raise two difficulties in the context of this study:

1.  Just as mentioned in the last point, we want to keep the expert judgment as unperturbed as possible. This means that we would need to create a prior that reproduces the 10-year and 1000-year uncertainty estimate of the experts, which would involve a complicated prior or an ad-hoc approach similar as currently described in the appendix A.
2.  A two-step approach of creating an EJ-prior and updating this with observations would in some cases make the outcome insensitive to the expert judgment. This is the same issue currently described in lines 196 – 205 and appendix B. So while it would remove this subjective component from the model, it would also tip the balance towards data and reduce the influence of expert judgments in the results.

We appreciate that considering EJ as priors would be more true to the Bayesian approach, but in the context of this study, which is defined around expert judgment, we'd rather keep EJ and data together in the likelihood of Eq. 6 such that we have the option to weigh them. Note that this point is further discussed in the response to the reviewer's line 173 comment.

3) given the procedure proposed, the tail of the distribution is controlled by expert judgement with a strength that is related to the subjective choice of the weight given to the expert "data" compared to the observed data. The result of the procedure is then assessed as credible/reasonable, but how could it not be so? From what I've understood, the procedure seems to allow a way to tweak subjectively the shape of the flood frequency distribution.

The weighing factor between data and EJ is indeed a modellers choice, needed to find a balance between data and EJ when the two don't 'align' nicely. We acknowledge that this is a subjective choice, or model choice, but so are many other model choices in this study, such as using a GEV-distribution. To substantiate this, we added the sensitivity analysis in appendix B

4) the results are not assessed against a benchmark. Why not using regional flood frequency analysis as a benchmark?

The benchmark of this study is the downstream discharge at Borgharen, used in two ways:

-   as historic observations (see Fig. 5)
-   as design flows from GRADE (as compared in Fig. 6)

Both add value, GRADE as it gives an estimate for large return periods, the observations as they are not a model result but measured data (and therefore cannot be judged for not 'foreseeing' the July 2021 event). The discharges at Borgharen are calculated from the experts' estimates for the discharges of the river tributaries, their dependence, and a sum-factor.

We'll make it clear to the reader that GRADE is a regional flood frequency analysis that includes historic events.

5) some of the methodological steps are unclear, sometimes, and should be properly explained (see the detailed comments below).

Detailed comments:

line 8: MCMC is just a tool. I would say here that you use Bayesian inference.

Thanks, will adopt this.

line 17: the 2021 peak at Borgharen is the highest but does not seem surprisingly high, looking at Figure 5. I think the same event has been much more surprising in other, smaller catchments. Even though it is surprising for the summer season, as I understand, your analysis later is not done accounting for seasonality. I would even expect that, if asked for the summer flood frequency curve, the experts would underestimate the probability of such an event.

That is right, it is more surprising in summer (the previous summer max was about 2000 $m^3$/s), and as well for the smaller contribution catchments. We indeed chose to not distinguish seasonality, as it would double the number of estimates which we think would have been too much.

line 30: here the text suggests that hydrological model simulations outperform statistical methods in flood frequency analysis. Has this been demonstrated in the literature? As far as I know, statistical models tailored for flood frequency analysis are more accurate than other methods both in gauged and ungauged basins (see Bloeschl et al., 2013, ISBN:9781107028180). Besides, despite some advantages, you clearly show limitations for the hydrological modelling approach in the discussion until line 45. Since the accurate estimate of the distribution tails is of interest, why don't you mention regional flood frequency analysis and inclusion of historical events as ways of increasing the robustness (and reducing the uncertainty) of the estimates? Besides, aren't design flows available from a regional frequency analysis in the area, e.g. to be used as a benchmark?

The GRADE model is the prevailing model that provides design flows for flood defence design. That is why the study's results are compared to it in Section 4.4. The black distributions in Figure 6 named "WBI-statistics" are from GRADE. The legend label should be changed to GRADE to avoid confusion, and we'll make sure that it is clear to the reader that the benchmark is a) GRADE, as b) the historical observations at Borgharen.

Furthermore, we did not mean to suggest that hydrological model simulations generally outperform statistical methods and will reword this. Note that GRADE is a combination of a statistical method and hydrological model, which might not have added to the clarity here.

line 43: I don't get the factor 3 vs. 1.4 sentence. What is the "outcome"?

We'll specify the quantities.

lines 65-68: spoiler alert! I would move this sentence after the results section.

We will (re)move this.

line 79: I don't get the meaning of the sentence "The discharge estimates for this catchment are therefore only used for expert calibration, as the flow is part of the French Meuse flow".

In this study, we modelled the overall catchment as a number of sub-catchments that flow into a main branch. The Semois sub-catchment is part of the larger French Meuse sub-catchment, i.e., it flows into the French Meuse tributary before this tributary enters the main branch. Therefore, it's not part of the sum-model (Eq. 1), as we would be double counting discharge. It is however a sub-catchment with a significant size and good data, which is why we did use it for expert calibration (i.e., comparing expert 10-year estimates to data).

We will clarify this or move the sentence down after were the model is introduced, where it will make more sense.

line 85: I would add a table here in the main text summarizing the data provided to the experts.

Will adopt this.

line 107: not having some more details on the construction of the correlation matrices is a pity. It would have been wise to publish that paper first.

That's right, but the other way around would be so as well. The two are related, too big to publish together, and trying to time them together is difficult with external factors. We found this order the most logical (bigger picture first, then zooming in on the details of dependence models and elicitation).

line 109: Each variable is modelled by a marginal distribution, it is not a distribution.

Will remove the "univariate" and take "marginal" out of the brackets.

Section 3.2: I am not sure that the ability of the expert in providing a judgement on the flood frequency curve can be measured by her/his ability in guessing the 10-yr flood in absolute terms. If you want the expert to help in reducing uncertainty in the tails of the distribution, she/he should inform you on how large floods may compare to small floods, by reasoning on the driving processes. In the end, it is the shape of the flood frequency distribution that's hard to get with local data, not the location. Your ranking seems to me tailored for getting the order of magnitude right, but not how surprising can large extreme events be compared to the more frequent ones. I know this cannot be done now but I would have asked the experts to guess the ratios between the 10-yr event and the mean event, and between the 100-yr event and 10-yr event, and so on, in order to get their perception on the shape of the distribution. Maybe you could discuss the idea in the discussion section, if you see that fit.

As discussed in point 1. We will add something in the discussion on this. From a catchment hydrology point of view it would perhaps make more sense to estimate ratios, as different runoff processes become less or more important. However, from a statistical and expert judgment point of view, were we want to quantify a statistical model for extreme river discharges, it makes most sense to try and estimate the components in that model more directly.

line 151: "a training exercise"

Will be corrected.

line 154: are the 26 questions made available somewhere?

These will be added to the supplementary information.

line 173: the weakly informed prior in Appendix A is very peculiar to me. I imagine very strange parameter combinations, very far from what could be expected for floods, are given the same weight than more reasonable ones, and some reasonable ones are excluded because of the bound at 10000. Why not the usual priors for the GEV distribution when dealing with floods, i.e., unbounded uniform for location and for the log of the scale and the Martins and Stedinger (2000, doi:10.1029/1999WR900330) geophysical prior, or similar ones, for the shape parameter?

The initial reason for choosing the weakly informed prior was that an uninformed prior did not result in uninformed 10-year and 1000-year estimates, which is what we were looking for. It gave bad results for the EJ-only results, as the 10-year and 1000-year distribution leave too much freedom for the uninformed GEV-parameter space. Aiming for uninformed 1000-year estimates evolved the approach to the one described in appendix A, in which a prior estimate for the shape parameter and a custom 'empirical' copula were used.

While the approach works and gives the required results, after reading this review, we consider exchanging the prior for the shape-parameter to the beta(6, 9) between [-0.5, 0.5] from Martins and Stedinger (2000). A quick analysis shows that it constrains the parameter space enough to give good results for the expert-only data, as shown in the figure below. This result is similar to Expert A, French Meuse in the supplementary material. Doing so will remove the need for appendix A, and make the approach less ad-hoc.

[Figure]

lines 185-195: here the expert information is accounted for as data (part of the likelihood). Why not accounting for it as prior information? That would be the natural way to do it: since the experts give their estimates without using discharge data, this can be considered prior information. For getting the prior distribution of the parameters from the prior assessment of the quantiles, one could use the procedure described in Renard et al. (2006, doi:10.1007/s00477-006-0047-4), for example. This would avoid the subjective choice of weights presented in lines 196-205, which actually control the fit of the tail of the flood frequency curves. Also, this would provide a more defendable prior than the one discussed in Appendix A.

As discussed in item 2 in the beginning of this document: considering the expert estimates as prior information and updating them with data, makes that the fitted distribution tend to follow the data more than the expert judgment. From a Bayesian perspective this is what you'd want, however it suppresses the effect of the expert's judgements in this EJ study. For that reason, we rather have the option of specifying the ratio between the two (as Eq. 6 provides).

This issue can also be viewed from the choice for the GEV distribution. The measured flows from the individual catchments could be considered independent identically distributed (iid). Not completely, due to climate change, and catchment changes, and human interactions (compromising the identically). But when combined with the expert estimates, this is further compromised. Therefore, instead of choosing a factor, we could have considered using a different or a mixed distribution that naturally fits the combination of data and estimates (but which?). We'll add this to the discussion.

line 196: log-likelihoods are summed

Indeed, will be corrected.

line 206: please indicate in which equation (and with what symbol) the "factor between the tributaries' sum and the downstream discharge" has been introduced. Is it the one in Eq. (1)? And what are the observations to which a log-normal distribution is fitted? I am confused here.

$f_{\Delta t,u}$, in Eq 1. We will add this. For the data-only model, an estimate for this factor was needed as well. For this, the historical factors were calculated and a log-normal distribution was used to parametrize this (it fitted well and is non-negative).

Section 3.4: I am sorry but I don't understand the procedure at all. I wish I could suggest how to improve points 1 and 2, but I can't figure out what they do mean.

We will clarify this. The main catch here is that (in our model) the dependence model does not only determine the strength with which each tributary's GEV-realization correlate during an event, but also how the GEV-distribution (within their tributary's bandwidths) are correlated.

If we would not have a bandwidth, but just a single GEV-parameter combination per tributary, it would be relatively easy: draw a [9 x 10000] sample of multivariate normal realizations and convert this to the GEV-space. Sum the discharges and multiply with 10000 realizations of the factor. This gives 10000 downstream discharges from which the

exceedance probabilities of interest can be calculated. (Note that this approach is repeated 2000 times)

We do however have a bandwidth of tributary discharge statistics from the MCMC trace, and would like to process this into the final answer. The bandwidth in the tail is mainly the cause of not having a lot of realizations in the tail (i.e., not having thousands years of data) We can reason that if one tributary would have a GEV-combination that gives a curve on the upper end of the bandwidth, it would be likely because of a high discharge event that its neighbour has experienced as well. The neighbouring tributary would therefore likely also have an 'upper' realization. This effect is modelled by correlating the GEV-realizations with the dependence model as well. So we draw a [9 x 2000] sample (9 tributaries, 2000 curves were generated), and transform these from normal space, to uniform space (cdf), to uniform [1, 8000] space. This gives an index of the GEV-parameter combination to choose (the GEV-curves will cross, so they were ordered based on their 1000-year discharge).

Lines 269-278: here it seems evident to me that the objective assigned to the expert is to guess a reasonable mean annual peak discharge, in m^3/s, but not so much the shape of the growth curve. Afterwards, the Cook's method values the experts in how well they get the order of magnitude of flood discharges right, more than the shape of the distribution. Is this what we need to inform our analysis about how extreme can large floods be?

The lines 269-278 describe the different approaches that experts chose to structure their reasoning. We tried to steer the experts as little as possible in choosing their approach (while you could argue that providing a certain type of data already does). All tried to structure their thinking in some way, some found an approach as calculating a mean annual peak discharge (or 10-year flow in our case) apparently most suitable, but we did not assign them a method.

Finally, it comes down to trusting that an expert's performance in estimating a 10-year flow gives a good indication of the expert's ability to estimate a 1000-year flow (and building on this trust by processing the results and comparing it with the Borgharen benchmark). There was no distinction between the importance of the return periods (such as, calculate mean annual flood and scale this to an extreme, would be). You can argue that estimating a growth-factor focusses the experts' attention more on the changes in catchment hydrology (or how big an extreme flood can be), but we prefer to directly assess the variable of interest (the 1000-year discharge) as it gives less ambiguity in processing the answers.

Line 290: "not too steep"

Will be corrected.

Figure 5: if I have understood well, the points in the third column should all be grey because discharges at Borgharen are not used in the fit. Am I right?

They should indeed be grey, thanks for the suggestion.

Line 308: I don't get what the following sentence means: "Sampling from these wide uncertainty bounds will therefore (too) often result in a high discharge event".

Will clarify or remove (perhaps this information is not too critical). What it means: When only fitting to data, the marginals will have very large uncertainty in the tail. Some sampled

tributary discharges will have extreme (e.g., >5000 m$^3$/s) discharges, even before combining with the other tributaries. With 9 sampled tributaries, there is a reasonable change that a combination has one of these samples, causing the end-result to be pushed up. Nothing too strange, just what we found when looking into the results.

Figure 6bc: it seems peculiar that combining the pieces of information that individually result in the blue and yellow distributions leads to the red one (e.g., the red mode is lower than the blue and yellow ones). Can you comment on that?

The difference is that the yellow results used EJ for both 10 and 1000-year, while the red only for 1000-year. The difference in mode order is due to the curves tilting a bit because of the 10-year EJ estimate (more clear in Figure 5ghi).

line 323: why are the median values considered best estimates?

50% chance it is lower, 50% chance it is higher. But this ambiguity is why it is between apostrophes, the most likely value could be seen as 'best estimate' as well. We'll remove it, a word like median probably doesn't need clarification for the readers of this article.

line 330: I don't understand the sentence.

"it can point out the statistically accurate experts to improve the extrapolated range": This means that the data can be used (in Cooke's method) to determine expert performance, assign them scores, and use these scores for to adjust the extrapolated range (in comparison to what it would have been without EJ). We'll replace 'improve' with something more neutral like 'adjust'.

line 340: but the experts knew about the 2021 event when doing the exercise and this has biased their estimates, I guess. How would have their estimates been different before 2021? That's hard to tell.

Probably lower as well, given that it is generally perceived as an unexpectedly large event. But then again, mainly for the summer season which was not considered separately in the estimates.

line 350: the following sentence doesn't mean anything to me: "were combined ... in ranges that are commonly 'in sample'".

Will remove "in ranges that are commonly 'in sample'"." part (it seems that it should not be there).

line 360: since the tails of the distributions are controlled by the expert opinions, it seems to me obvious that they "seem credible". Couldn't they be compared to the outcomes of a more classical regional flood frequency analysis?

The tails are compared to the discharge at Borgharen, which is calculated from their estimates (and can be validation reasonably well because of the long data record). So while this is obvious for the tributary estimates, the final discharges at Borgharen (as shown in the supplementary information) do not have to seem credible.

---

## Author Comment (AC2)

The authors would like to thank the reviewer for reviewing the article. We have written a response to the reviewer's comments below. If the reviewer agrees with our interpretation of the comments and the responses, we will process this as textual changes in the final version of the article (depending on how that proceeds).

This study investigates through the Cooke's method how scientific judgments by experts can assist flood-risk managers. In my opinion, there are several issues that need to be addressed so that the applied methods, justification, and results, can be clearer and of practical use to other case studies. Please see several such comments and suggestions below:

1) The main point raised by the authors for someone to use the suggested method is that "...existing statistical and hydrological models that estimate these discharges often lack transparency regarding the uncertainty of their predictions..."; however, please note that the purpose of the probabilistic analysis is exactly this one (i.e., to estimate and take into consideration the uncertainty and variability of predictions of the input and output parameters of a flood model; see for example, a review, applications, and discussion on the uncertainty of flood parameters through benchmark examples in Dimitriadis et al., 2016). I would suggest not comparing with such methods (which are plenty in the literature), but focusing on the advantages and limitations of the proposed method.

That is a good suggestion. We'll rephrase parts of the introduction such that it doesn't seem like a "physics-based modelling does not incorporate uncertainty" statement. Right now it does not rightly acknowledge the variety in different modelling approach, and their respective assessment of uncertainty (from which Dimitriadis et al. 2016) is an example).

Dimitriadis, P., A. Tegos, A. Oikonomou, V. Pagana, A. Koukouvinos, N. Mamassis, D. Koutsoyiannis, and A. Efstratiadis, Comparative evaluation of 1D and quasi-2D hydraulic models based on benchmark and real-world applications for uncertainty assessment in flood mapping, Journal of Hydrology, 534, 478–492, doi:10.1016/j.jhydrol.2016.01.020, 2016.

2) The fact that "...the devastating flood event that occurred in July 2021... was not captured by the existing model for estimating design discharges.", is not for the statistical methods to blame (or replace), but a more appropriate analysis by experts should have been performed. For example, there is an application shown in Figure 10 (in Dimitriadis et al., 2016), where there was a certain flooded area that could not be captured by a 1D model (due to the 1D nature of the model that cannot account for a 180 degrees turn of the water, since only 1 direction is possible within a cross-section), whereas this area can be captured if a 2D (or quasi-2D) model is applied. However, only an expert in flood modeling could identify this (e.g., the authors state that "The study demonstrates that utilizing hydrological experts in this manner can provide plausible results with a relatively limited effort, even in situations where measurements are scarce or unavailable."). If this is what the authors are trying to highlight in this work (i.e., that the flood models should not be blindly applied by non-experts), then this is a strong and important statement, which however needs to be further discussed.

Up to a certain point, the impacts of extreme events can be estimated better when a good analysis of the hydraulic details are made. For the July 2021 flood, a clear example of this is the effect of dams in the catchment (hydro-power as well as weirs). The main cause of the event being surprisingly large was however the meteorological situation. The most important point here is that practitioners need to be aware of the uncertainties in their modelling approach. We'll try and add this to the introductory text.

3) Please consider rephrasing the sentence "Quantifying events that are more extreme than ever measured (i.e., with return levels that are longer than the time period of representative measurements), requires extrapolating from available data or knowledge.", since it is not exactly true. The return period T corresponds to a probability of occurrence (i.e., on average, a storm event is expected to occur in T years) and not a deterministic occurrence that involves any kind of extrapolations or specific (i.e., 5th, 95th etc.) quantiles (please see the mathematical definitions and methods for extreme analysis and probability fitting in a recent work by Koutsoyiannis, 2022).

This is a good point. The current wording indeed suggest that historic data carry a deterministic return period that can be extrapolated like a data point, while this requires modelling assumptions (such as a probability distribution). We will rephrase this.

Koutsoyiannis, D., Replacing histogram with smooth empirical probability density function estimated by K-moments, Sci, 4 (4), 50, doi:10.3390/sci4040050, 2022.

4) The application of Cooke's method to the specific study is not very clear to me. For example, the authors state that "A simple statistical model was developed for the river basin, consisting of correlated GEV-distributions for discharges in upstream sub-catchments. The model was fitted to expert judgments, measurements, and the combination of both, using Markov chain Monte Carlo. Results from the model fitted only to measurements were accurate for more frequent events, but less certain for extreme events."; since they were all experts and applied the same model, how come they came up with different results, did they use different methods, and what are these methods? where did the experts base their reply, did they perform also simulations or just probabilistic fitting?

The model (Eq. 1), is a framework to process the experts' estimates. The experts had to come up with 10-year and 1000-year discharge estimates (5th, 50th and 95th percentiles, such that it is an uncertainty assessment). The experts were free in choosing their methods. It was however a 1-day expert session, in which the experts had to come up with uncertainty estimates for 10 tributaries, which tends to steer them towards using simpler 'models' for making their estimates. The experts didn't have to do simulations or probability fitting themselves. The model framework was discussed with them, and they had to fill in the numbers with which the estimates of extremes can be generated (through the model framework).

5) In my opinion, it is not very appropriate to apply a Monte-Carlo method with so few samples; please consider including more samples. Also, how come "The combined approach provided the most plausible results, with Cooke's method reducing the uncertainty by appointing most weight to two of the seven experts."; why the authors have selected these 2 scientists; were these two more experts than the other scientists?

We will clarify Section 3.4. It suggests that we used 10.000 samples for each tributary, but this process is repeated 2000 times. The 10.000 realizations for each tributary are used to generate an exceedance frequency curve for the downstream location. By doing this 2000 times, we also get uncertainty bounds for this. These 10.000 realizations represent 10.000 years, which is sufficient for estimating a 1000-year flow, given that the process is repeated 2000 times.

6) More details are required to back up the statement "The discharge at the Dutch border exceeded the flood events of 1926, 1993, and 1995. Contrary to those events, this flood occurred during summer, a season that is (or was) often considered irrelevant for extreme discharges on the Meuse."; please perform a proper statistical analysis and identify for each season the appropriate probability distribution to show at what discharge the probability of occurrence in the summer season exceeds the selected return period.

We will add some numbers from the (Force Fact-finding hoogwater, 2021) report to the article (already in the reference list): The corresponding author of this article did the EV-analysis for the discharges in that report, which showed that the flow had a 120-year average recurrence interval based on year-round statistics, and 610-year average recurrence interval when considering only the summer half year (April to September). These estimates are based on MCMC-fitted GEV-distributions including the 2021 event. Please refer to figure 2.5 in that report (unfortunately Dutch only).

7) Regarding the comments "The event was thus surprising in multiple ways. This might happen when we experience a new extreme, but given that Dutch flood risk has safety standards up to once per 100,000 years (Ministry of Infrastructure and Environment, 2016) one would have hoped this to be less of a surprise." and "While most studies aimed at obtaining better estimates of discharge extremes use hydrological or statistical modeling, some follow the approach of using expert judgment (EJ).", please note that this is a must point in every scientific application, since when non-experts apply methods they do not understand, it could lead to failure regardless the magnitude of the selected return period.

Agree, we'll add a comment that, while often not explicitly, all modelling involves (or at least should) expert judgment to some extent.

8) It is mentioned that "For the Dutch rivers Meuse and Rhine, the GRADE instrument is used for this. It generates 50,000 years of rainfall and discharges."; please give more details on this model and how it generates so long rainfall and discharge timeseries (does it use a stochastic simulation approach for the rainfall annual extremes and input these to a hydraulic model to produce the discharge at a specific location in the area of interest?).

The GRADE model is not scientifically published, but it is well described in this report: https://publications.deltares.nl/1209424_004_0018.pdf (Referred to as Hegnauer et al., 2014 in the article). We will add some more details to the article, but the reviewer's guess is right: It resamples the historically observed rainfall while preserving the spatial and temporal correlation. This rainfall is then processed through a hydrologic model to generate tributary discharges that are simulated with a hydraulic model.

---

## Author Response (AR1)

Dear Mr. Viviroli and reviewers,

Hereby we submit the updated version of the article: "Using structured expert judgment to estimate extremes: a case study of discharges in the Meuse River". We took the feedback of the reviewers and editor to heart and made significant changes to the article. The article now has a more general focus on estimating (hydrological) extremes rather than being specifically tailored to the Meuse's extreme discharges. While the latter is still the article's case-study, the following changes should make the research more appealing to a broader public of statistically focussed hydrologists:

- Cooke's method and structured expert judgment have been given a more proper introduction: How it compares to and formalizes regular expert judgment.

- The Bayesian approach is more general now (thanks to reviewer 1's feedback). The ad-hoc prior and the extra weight for EJ-likelihood are removed. This should make the approach more easily applicable (and therefore more relevant) for other studies.

- In line with this, the abstract, introduction, discussion, and conclusion now reflect on the study as a method to estimate 'out of sample' extremes in general, and less on the comparison between the study's results (i.e., using data, using EJ, or using both).

The next page contains an overview of the main changes made to the manuscript, ordered by section. A detailed response to both referees' comments is found on the pages thereafter. These responses have been updated from the response we had given during the open discussion phase.

Together with this document, we uploaded:

- The new version of the article.

- The new version of the supplementary information, now also including the questionnaire through which the uncertainties were elicited.

- A comparison between the old and new version using track changes. Note that large parts of the text have been changed, which makes it hard to track the differences. Therefore, we included line numbers in the response to the comments (the tables hereafter) to indicate specifically where the comments have been processed.

We think that the manuscript changes are a substantial improvement and hope that the new version will be reconsidered. However, regardless of the decision, we would like to thank the reviewers for their effort and input so far, as their feedback has greatly improved the presentation of our research.

Kind regards,

Guus Rongen
Oswaldo Morales-Nápoles
Matthijs Kok

**Overview of the main changes per section**

1. Introduction
   - GRADE is now clearly presented as the benchmark in this study and named as a regional flood frequency analysis (in response to referee 1's comments).
   - We are no longer comparing the presented 'data-based' approach to a 'model-based' approach (in response to referee 2's comments). This presents the choice as binary, while lots of hydrological models are a combination. We now present several approaches of extending a data record, and present expert judgment as an alternative to these.
   - Structured expert judgment is introduced more properly and is compared to regular 'everyday' expert judgment (in response to referee 2's comment). This should make the exact meaning of it more clear to a broader audience.
2. Study area and used data.
   - A list of used data is given (in response to referee 1's comments)
3. Method description
   - Section 3.2: Cooke's method is introduced more clearly. What it is, why you should use it, and specifically the *structured* part.
   - Section 3.3: This section is largely rewritten, to match it with the changes in the Bayesian approach (i.e., the geophysical prior and removing the EJ-factor). The explanation is more formal as well (i.e., includes a proper mathematical description).
   - Section 3.4 is simplified (in response to referee 1's comments) and is accompanied by a new appendix (A) that gives a mathematical description of the algorithmic steps used to sample downstream discharges.
4. Results
   - The result sections did not change significantly apart from some extra explanation on the downstream discharge results in Section 4.4.
5. Discussion
   - This section has been rewritten completely. It is split into two parts, method-related (5.1) and result-related (5.2). Section 5.1 explains why we elicit discharges instead of ratios or shape parameters (in response to referee 1's comments), the choice and suitability of the GEV distribution, and the omission of seasonality. Section 5.2 discusses the validity of results, it
     - reflects on bad GEV fits and bad Cooke's Method scores (which correlate),
     - reflects on the comparison with GRADE,
     - compares estimation of extremes through EJ (with Cooke's method) to extrapolation with a model, and
     - explains the EJ-only approach (without using data) as too uncertain.
6. Conclusions
   - The conclusions now focus less on comparing the combination of data and EJ to data-only and EJ-only, as that is less relevant for general applications.
   - The conclusions contain a general statement on how expert judgment can be used to limit uncertainty through a Bayesian approach, because this is an important 'take-away' message for the readers.

| # | Referee comment | Authors' response |
|---|---|---|
| **RESPONSE TO REFEREE 1'S COMMENTS** | | |
| **#** | **Referee comment** | **Authors' response** |
| **1** | 1) I am not sure that the ability of the expert in providing a judgement on the flood frequency curve can be measured by her/his ability in guessing the 10-yr flood in absolute terms. If one wants the expert to help in reducing uncertainty in the tails of the distribution, she/he should inform us on how large floods may compare to small floods, by reasoning on the driving processes. In the end, it is the shape of the flood frequency distribution that's hard to get with local data, not the location. The proposed method seems to be tailored for getting the order of magnitude right, i.e., the flood magnitude in m^3/s, but not how surprising can large extreme events be compared to the more frequent ones. | The proposed method is indeed tailored to get the flood magnitude right. However, we do so by applying Cooke's method for Structured expert judgment, in which the experts are scored based on their ability to estimate the 10-year discharges. These are then used to estimate the 1000-year discharge. Consequently, experts that score high will have a 10-year estimate that corresponds to the observations, and a defendable estimate for the 1,000-year. Together, these indicate the ratio, or shape (if combined with data) between less extreme and more extreme discharges. We chose to elicit discharges rather than a ratio or shape parameter, as it directly informs our quantity of interest. Indirectly, these parameters are thus derived from the 10-year and 1000-year estimates. Eliciting such parameters directly could indeed change the focus more to a comparison between events of different extremity, which is now discussed in Sect. 5.1, lines 445 – 459. |
| **2** | 2) the expert information is accounted for as data (part of the likelihood) using an ad-hoc procedure, which seems to me inconsistent with the Bayesian way. Why not accounting for expert judgement as prior information? That would be the natural Bayesian way to do it: since the experts give their estimates without using discharge data, this can be considered as prior information. | We thank the reviewer for this suggestion, which has greatly improved the presentation of this research. We have changed the approach for Bayesian inference in three ways (see article Sect. 3.3):

- A geophysical prior is used for the shape parameter. This replaces the ad-hoc prior from the old Appendix A, making the approach simpler and more defendable.
- The expert estimates are considered priors now (additional to the geophysical shape-parameter prior). This is mainly a matter of wording, the contribution to the posterior likelihood is unchanged, except that:
- The weighing factor between expert judgment and observations is removed (including Appendix B, which showed its sensitivity). Rather than trying to fix deviating expert estimates, we now use it as an extra check if the experts' estimates are plausible (Experts D and E, with high scores, have 1000-year estimates that align with the observations, through the GEV).

Regarding this specific comment: we do now consider the expert estimates a-priori information (see lines 215, 532). |

| 3 | 3) given the procedure proposed, the tail of the distribution is controlled by expert judgement with a strength that is related to the subjective choice of the weight given to the expert "data" compared to the observed data. The result of the procedure is then assessed as credible/reasonable, but how could it not be so? From what I've understood, the procedure seems to allow a way to tweak subjectively the shape of the flood frequency distribution. | As described under the response to the last comment (2), we have removed the factor. Estimates that do not match the observations through the GEV-estimates can in some cases be defendable. For example, when an extreme would no longer be considered to be from the same population (i.e., identically distributed) compared to less extreme events. However, the GEV is generally flexible enough to facilitate light and heavy tails. So, rather than fixing deviating expert estimates, we accept them and use them as a check of whether the high-scoring experts have estimates matching the GEV. This is the case, which also means that 'bad' fits are filtered out through Cooke's method.

See the discussion on using the downstream discharges as check in lines 438-444, and the further discussion on the GEV as validation in line 460-469, and 485-489. Throughout the text, we have clarified that we estimate upstream discharges, and calculate downstream discharges. |
| --- | --- | --- |
| 4 | 4) the results are not assessed against a benchmark. Why not using regional flood frequency analysis as a benchmark? | We have added some more explanation on GRADE. This model (Generator of Rainfall And Discharge Extremes) a variant on a conventional regional flood frequency analysis that includes historic events. It resamples historical rainfall and simulates this with hydrological models (instead of estimating discharges from statistical catchment properties). We clarify in the latest version of the paper that our results are assessed against this benchmark:

Line 36-39 introduce GRADE as regional flood frequency analysis. Line 82 explains the comparison to GRADE (benchmark), which is further presented in line 414 onwards. GRADE's suitability as comparison is discussed in line 490-499 |
| | 5) some of the methodological steps are unclear, sometimes, and should be properly explained (see the detailed comments below). | See below |
| 5 | line 8: MCMC is just a tool. I would say here that you use Bayesian inference. | Changed (line 8) |
| 6 | line 17: the 2021 peak at Borgharen is the highest but does not seem surprisingly high, looking at Figure 5. I think the same event has been much more surprising in other, smaller catchments. Even though it is surprising for the summer season, as I understand, your analysis later is not done accounting for seasonality. I would even expect that, if asked for the summer flood frequency curve, the experts would underestimate the probability of such an event. | That is right, it is more surprising in summer (the previous summer max was about 2000 m3/s), and as well for the smaller contribution catchments. We indeed chose to not distinguish seasonality, as it would double the number of estimates which we think would have been too much. This methodological choice is now discussed in the discussion (Sect. 5.1, lines 469-474). |

| 7 | line 30: here the text suggests that hydrological model simulations outperform statistical methods in flood frequency analysis. Has this been demonstrated in the literature? **@1** As far as I know, statistical models tailored for flood frequency analysis are more accurate than other methods both in gauged and ungauged basins (see Bloeschl et al., 2013, ISBN:9781107028180). Besides, despite some advantages, you clearly show limitations for the hydrological modelling approach in the discussion until line 45. Since the accurate estimate of the distribution tails is of interest, why don't you mention regional flood frequency analysis and inclusion of historical events as ways of increasing the robustness (and reducing the uncertainty) of the estimates? Besides, aren't design flows available from a regional frequency analysis in the area, e.g. to be used as a benchmark? **@2** | **@1**: Not as far as we know. We have removed the part in the introduction that suggests it (which we did not mean to do) and made it less "models vs. statistics". **@2**: Like mentioned in the answer to comment 4, we now present GRADE as (a variant to) a regional flood frequency analysis, which it is. We have clarified that the results are compared to this (next to observed discharges), and Fig 7 (previously Fig. 6) now mentions GRADE as well, instead of "WBI-statistics", which is the same but was not introduced. |
|---|---|---|
| 8 | line 43: I don't get the factor 3 vs. 1.4 sentence. What is the "outcome"? | We have removed this sentence and the reference. |
| 9 | lines 65-68: spoiler alert! I would move this sentence after the results section. | We have removed this paragraph |
| 10 | line 79: I don't get the meaning of the sentence "The discharge estimates for this catchment are therefore only used for expert calibration, as the flow is part of the French Meuse flow". | In this study, we modelled the overall catchment as a number of sub-catchments that flow into a main branch. The Semois sub-catchment is part of the larger French Meuse sub-catchment, i.e., it flows into the French Meuse tributary before this tributary enters the main branch. Therefore, it's not part of the sum-model (Eq. 1), as we would be double counting discharge. It is however a sub-catchment with a significant size and good data, which is why we did use it for expert calibration (i.e., comparing experts' 10-year estimates to data). We have moved the sentence down to the method section that explains the sampling method (line 301-302). Here it becomes relevant (why we sample from 9 instead of 10 tributaries). The reader should at this point have enough background to better understand it. |
| 11 | line 85: I would add a table here in the main text summarizing the data provided to the experts. | We have added a list with a short description of the provided data in Sect. 2 (lines 103-115). |
| 12 | line 107: not having some more details on the construction of the correlation matrices is a pity. It would have been wise to publish that paper first. | We agree with the reviewer's observation. However, publishing the dependence results before these would render similar problems. The two are related, too big to publish together, and trying to time them together is difficult with external factors. We found this order the most logical (bigger picture first, then zooming in on the details of dependence models and elicitation). |
| 13 | line 109: Each variable is modelled by a marginal distribution, it is not a distribution. | This line is no longer present in the revised article. |

| 14 | Section 3.2: I am not sure that the ability of the expert in providing a judgement on the flood frequency curve can be measured by her/his ability in guessing the 10-yr flood in absolute terms. If you want the expert to help in reducing uncertainty in the tails of the distribution, she/he should inform you on how large floods may compare to small floods, by reasoning on the driving processes. In the end, it is the shape of the flood frequency distribution that's hard to get with local data, not the location. Your ranking seems to me tailored for getting the order of magnitude right, but not how surprising can large extreme events be compared to the more frequent ones. I know this cannot be done now but I would have asked the experts to guess the ratios between the 10-yr event and the mean event, and between the 100-yr event and 10-yr event, and so on, in order to get their perception on the shape of the distribution. Maybe you could discuss the idea in the discussion section, if you see that fit. | Please refer to comment 1 on why we chose to elicit 10-year and 1000-year discharges. Regarding the discussion on assessing the weight of an expert based on their ability to estimate a ratio rather than an absolute value: that is a valid point. It would indeed be interesting to compare the results with a study focused on that. We have added this to the discussion (lines 445 – 459). |
|----|----|----|
| 15 | line 151: "a training exercise" | Corrected (line 201) |
| 16 | line 154: are the 26 questions made available somewhere? | They have been added to the supplementary material |
| 17 | line 173: the weakly informed prior in Appendix A is very peculiar to me. I imagine very strange parameter combinations, very far from what could be expected for floods, are given the same weight than more reasonable ones, and some reasonable ones are excluded because of the bound at 10000. Why not the usual priors for the GEV distribution when dealing with floods, i.e., unbounded uniform for location and for the log of the scale and the Martins and Stedinger (2000, doi:10.1029/1999WR900330) geophysical prior, or similar ones, for the shape parameter? | As mentioned in the response to comment 2, we have changed the old weakly informed prior by the geophysical prior from Martins and Stedinger (2000, doi:10.1029/1999WR900330). See lines 239 – 248. It is indeed a much more straightforward way of limiting the shape-variability of the GEV. For the location parameter, we used a weakly informed prior that gives a uniform likelihood for all positive values (-inf for negative flows which can safely be assumed infeasible in our area of application). For the scale parameter we used a uniform distribution for positive values as well. Note that we did not use the Jeffrey's prior (i.e., 1/scale uniform), as this gave bad results in combination with the expert estimates and without data: The very high probability density of scale values close to 0 would result in a more or less horizontal GEV curve: a discharge for the location parameter that seems plausible given the 10-y and 1000-y expert estimates, and a near-zero scale. The prior is discussed in lines 237-252. |

| 18 | lines 185-195: here the expert information is accounted for as data (part of the likelihood). Why not accounting for it as prior information? That would be the natural way to do it: since the experts give their estimates without using discharge data, this can be considered prior information. For getting the prior distribution of the parameters from the prior assessment of the quantiles, one could use the procedure described in Renard et al. (2006, doi:10.1007/s00477-006-0047-4), for example. This would avoid the subjective choice of weights presented in lines 196-205, which actually control the fit of the tail of the flood frequency curves. Also, this would provide a more defendable prior than the one discussed in Appendix A. | As mentioned in the response to comment 2, we now consider expert estimates to be prior information. We incorporate this into the posterior log-likelihood function with the method presented in Viglione et al. 2013: DOI:10.1029/2011WR010782. We find this a straightforward and easy to implement procedure for incorporating expert judgment. Note that the subjective choice of weights, previously discussed in 196-205, was not needed because of this procedure. It was due to the likelihood of the observations being dominant compared to the prior likelihood. We suspect (but haven't checked this), that this would be similar when following a different procedure such as mentioned by the reviewer. |
|---|---|---|
| 19 | line 196: log-likelihoods are summed | Corrected (line 266). |
| 20 | line 206: please indicate in which equation (and with what symbol) the "factor between the tributaries' sum and the downstream discharge" has been introduced. Is it the one in Eq. (1)? And what are the observations to which a log-normal distribution is fitted? I am confused here. | $f_{\Delta t}$, in Eq. 1. We have added this (line 274-275). For the data-only model, an estimate for this factor was needed as well. For this, the historical factors were calculated, and a log-normal distribution was used to parametrize this (it fitted well and is non-negative). This text in lines 274-286 was adjusted to make this procedure clearer). |
| 21 | Section 3.4: I am sorry but I don't understand the procedure at all. I wish I could suggest how to improve points 1 and 2, but I can't figure out what they do mean. | We have clarified this procedure. Section 3.4 now contains a conceptual explanation, and (the new) Appendix A has been added to give a step-by-step overview of the calculation. Together, we trust the method will be clearer to the reader. |
| 22 | Lines 269-278: here it seems evident to me that the objective assigned to the expert is to guess a reasonable mean annual peak discharge, in m^3/s, but not so much the shape of the growth curve. Afterwards, the Cook's method values the experts in how well they get the order of magnitude of flood discharges right, more than the shape of the distribution. Is this what we need to inform our analysis about how extreme can large floods be? | Please refer to the responses on comment 1 and 14. We appreciate your suggestion of estimating ratios and assessing experts by their ability of estimating ratios, which we've added to the discussion. |
| 23 | Line 290: "not too steep" | Corrected (line 383) |
| 24 | Figure 5: if I have understood well, the points in the third column should all be grey because discharges at Borgharen are not used in the fit. Am I right? | We have made these grey, as well as the observation dots in the supplementary information for the Borgharen discharges. (See Figure 6 (prev. 5) and the supplementary material) |

| 25 | Line 308: I don't get what the following sentence means: "Sampling from these wide uncertainty bounds will therefore (too) often result in a high discharge event". | To give more background on this: when usually fitting a model, the whole model is fitted to the end result. For the sake of the expert elicitation, we fitted it to individual components, just like the expert were estimating. When sampling from these components, the model-simplifications (e.g., the Gaussian copula does not exactly reproduce tail dependence, or the fitted GEV does not exactly reproduce the tributary discharges) result in a slight deviation. |
|----|------|------|
| 26 | Figure 6bc: it seems peculiar that combining the pieces of information that individually result in the blue and yellow distributions leads to the red one (e.g., the red mode is lower than the blue and yellow ones). Can you comment on that? | This is because the red line results from the GL DM, and the yellow line from the EQ DM. The latter has a higher 'factor between upstream sum and downstream'. This is now explicitly explained in lines 402-405. |
| 27 | line 323: why are the median values considered best estimates? | The term 'best estimates' is removed. Median should suffice for the readers. (Line 417) |
| 28 | line 330: I don't understand the sentence. | The sentence is changed for: "Including expert estimates, weighted by their ability to estimate the 10-year discharges, improves the precision of discharge estimates in the range of extremes." Note that while Cooke's method gives a defendable way of saying that the *accuracy* improves, we stick here to the safer statement of saying the *precision* (i.e., the narrowness of the uncertainty intervals) improves (line 433). |
| 29 | line 340: but the experts knew about the 2021 event when doing the exercise and this has biased their estimates, I guess. How would have their estimates been different before 2021? That's hard to tell. | Indeed, it most certainly did affect their estimates, and it would most likely have affected the comparison to GRADE as well (if GRADE would have included the 2021 event). This is now more elaborately discussed in the comparison to GRADE in the discussion (lines 490-499 and specifically line 497). |
| 30 | line 350: the following sentence doesn't mean anything to me: "were combined ... in ranges that are commonly 'in sample'". | Changed for: "Experts' estimates of tributary discharges during a once per 10 year and once per 1,000-year event are combined with high river discharges measured over the past 30-70 years." (Lines 518 – 520) |
| 31 | line 360: since the tails of the distributions are controlled by the expert opinions, it seems to me obvious that they "seem credible". Couldn't they be compared to the outcomes of a more classical regional flood frequency analysis? | Please refer to the response to comment 4, GRADE can be considered a proper regional flood frequency analysis. Moreover, the expert did not estimate the downstream discharges directly, so their knowledge of these discharges could not directly inform their estimates (regarding tributary discharges). |

| RESPONSE TO REFEREE 2'S COMMENTS | | |
|---|---|---|
| **#** | **Referee comment** | **Authors' response** |
| **1** | 1) The main point raised by the authors for someone to use the suggested method is that "...existing statistical and hydrological models that estimate these discharges often lack transparency regarding the uncertainty of their predictions..."; however, please note that the purpose of the probabilistic analysis is exactly this one (i.e., to estimate and take into consideration the uncertainty and variability of predictions of the input and output parameters of a flood model; see for example, a review, applications, and discussion on the uncertainty of flood parameters through benchmark examples in Dimitriadis et al., 2016). I would suggest not comparing with such methods (which are plenty in the literature), but focusing on the advantages and limitations of the proposed method. | We have rephrased parts of the introduction to remove to the model-based versus statistics-based suggestion. Like suggested by the reviewer, we focus on the advantages and limitations of using structured expert judgment, to reduce the uncertainty in extreme discharges. Some other approaches of doing this (e.g., paleoflood, historical archives) are presented as a comparison in the introduction now (lines 51-55), rather than a comparison to model-based approaches. The (mostly new) discussion Sect. 5.1 discusses the advantages and limitations of the proposed method. |
| **2** | 2) The fact that "...the devastating flood event that occurred in July 2021... was not captured by the existing model for estimating design discharges.", is not for the statistical methods to blame (or replace), but a more appropriate analysis by experts should have been performed. For example, there is an application shown in Figure 10 (in Dimitriadis et al., 2016), where there was a certain flooded area that could not be captured by a 1D model (due to the 1D nature of the model that cannot account for a 180 degrees turn of the water, since only 1 direction is possible within a cross-section), whereas this area can be captured if a 2D (or quasi-2D) model is applied. However, only an expert in flood modeling could identify this (e.g., the authors state that "The study demonstrates that utilizing hydrological experts in this manner can provide plausible results with a relatively limited effort, even in situations where measurements are scarce or unavailable."). If this is what the authors are trying to highlight in this work (i.e., that the flood models should not be blindly applied by non-experts), then this is a strong and important statement, which however needs to be further discussed. | We agree with the reviewer that up to a certain point, the impacts of extreme events can be estimated better when a good analysis of the hydraulic details are made. For the July 2021 flood, a clear example of this is the effect of dams in the catchment (hydropower as well as weirs). The main cause of the event being surprisingly large was however the meteorological situation. Therefore, we do not think an appropriate analysis of the hydraulics would have led to the model (GRADE) capturing the event.

The most important point here is that practitioners need to be aware of the uncertainties in their modelling approach, a point on which we think the reviewer and the authors agree, if we understand your comment correctly. Accordingly, we do now clearly present 'expert judgment' as the ability to use one's experience to verify observations (referring to the reviewer's "a more appropriate analysis by experts should have been performed"). And structured expert judgment with Cooke's method as way of formalizing expert judgment. Lines 56-60 introduce expert judgment in this context, and the discussion (both 5.1 and 5.2) now more extensively contains a description of how the participants performed in this regard. |

| 3 | 3) Please consider rephrasing the sentence "Quantifying events that are more extreme than ever measured (i.e., with return levels that are longer than the time period of representative measurements), requires extrapolating from available data or knowledge.", since it is not exactly true. The return period T corresponds to a probability of occurrence (i.e., on average, a storm event is expected to occur in T years) and not a deterministic occurrence that involves any kind of extrapolations or specific (i.e., 5th, 95th etc.) quantiles (please see the mathematical definitions and methods for extreme analysis and probability fitting in a recent work by Koutsoyiannis, 2022). | The previous wording indeed suggest that historic data carry a deterministic return period that can be extrapolated like a data point. We have changed this for: " Estimating the magnitude of events greater than the largest from historical (representative) records is a nontrivial task. It requires establishing a model that describes the occurrence of such events and subsequently extrapolating to specific exceedance probabilities from this model." (Line 30-31) |
|---|---|---|
| 4 | 4) The application of Cooke's method to the specific study is not very clear to me. For example, the authors state that "A simple statistical model was developed for the river basin, consisting of correlated GEV-distributions for discharges in upstream sub-catchments. The model was fitted to expert judgments, measurements, and the combination of both, using Markov chain Monte Carlo. Results from the model fitted only to measurements were accurate for more frequent events, but less certain for extreme events."; since they were all experts and applied the same model, how come they came up with different results, did they use different methods, and what are these methods? where did the experts base their reply, did they perform also simulations or just probabilistic fitting? | The participants in the study are all called experts in the article. They were pre-selected on their field of expertise (practitioners or researchers in hydrology). However, their expertise as uncertainty assessors is subsequently assessed using Cooke's method. Therefore, whether they are expert (in assessing uncertainty) in the context of this study is determined based on their estimates for the 10-year discharges (explained in Sect. 3.2, and explicitly in line 207-208). We have clarified this by giving Cooke's method a more proper introduction (line 56-73, and Sect. 3.2).

Note that the model (Eq. 1), is a framework used by us to process the experts' estimates. The experts had to come up with 10-year and 1000-year discharge estimates (5th, 50th and 95th percentiles, such that it is an uncertainty assessment). The experts were free in choosing their methods to come up with the estimates needed to fill in Eq. 1. This was indeed unclear, so to clarify this, the first sentence of Sect. 3 is now (note the bold words that were added): *To obtain estimates for downstream discharge extremes, experts needed to quantify **different components in a simple model that gives the downstream discharge as the sum of the tributary discharges, times a factor correcting for covered area and hydrodynamics.***

The expert session was a 1-day expert session (lines 200-203), in which the experts had to come up with uncertainty estimates for 10 tributaries, which tends to steer them towards using simpler 'models' for making their estimates. The experts didn't have to do simulations or probability fitting but could so if they deemed it necessary. We have added this in Sect. 4.2. as well (lines 368-370). |

| 5 | 5) In my opinion, it is not very appropriate to apply a Monte-Carlo method with so few samples; please consider including more samples. Also, how come "The combined approach provided the most plausible results, with Cooke's method reducing the uncertainty by appointing most weight to two of the seven experts."; why the authors have selected these 2 scientists; were these two more experts than the other scientists? | We have clarified Section 3.4. It is now split in a more conceptual part (Sect. 3.4) and a more mathematical part (Appendix A). We used 10,000 samples for each tributary, which are used to generate an exceedance frequency curve for the downstream location. These 10,000 yearly discharges are sufficient to cover up to the 1,000-year range. By doing this 10,000 times, we also get uncertainty bounds for this (10,000 was deemed sufficient for estimating the 2.5th, 35th, 50th, 75th, and 97.5th percentiles). We now clearly mention that the whole simulation comprises 100,000,000 samples (lines 306, 576), but split in 10,000 times 10,000 to create uncertainty bounds.

Regarding the second part of the reviewer's question: The 2 experts were assigned the greatest weights based on their statistically accurate estimates for the 10-year discharges on the 10 considered tributaries. Within the context of this study (lines 207-208), we consider their uncertainty estimates more valuable. Saying that they are more experts than the others would have a connotation we wish to avoid. To clarify this, the sentence in the abstract (lines 10-11) was changed to: "Cooke's method reduced the uncertainty by appointing most weight to the two experts that could most accurately estimate more frequent discharges." |
| 6 | 6) More details are required to back up the statement "The discharge at the Dutch border exceeded the flood events of 1926, 1993, and 1995. Contrary to those events, this flood occurred during summer, a season that is (or was) often considered irrelevant for extreme discharges on the Meuse."; please perform a proper statistical analysis and identify for each season the appropriate probability distribution to show at what discharge the probability of occurrence in the summer season exceeds the selected return period. | We added the exceedance frequencies presented in the (Force Fact-finding hoogwater, 2021) report to the article (already in the reference list): The corresponding author of this article did the EV-analysis for the discharges in that report, which showed that the flow had a 120-year average recurrence interval based on year-round statistics, and 600-year average recurrence interval when considering only the summer half year (April to September). Please refer to lines 22-25. These estimates are based on MCMC-fitted GEV-distributions including the 2021 event. Please refer to figure 2.5 in that Dutch report.

We acknowledge that this is a Dutch report. An international publication closely related to this report is on its way but has not been published yet. If the reviewer would find it necessary, we see if we could include the EV-analysis in that report (which was done in a similar manner as the 'data-only' approach in this study) as an appendix in this article. We'd have to discuss this with the authors of the just mentioned yet to be published article to avoid duplication. |

| 7 | 7) Regarding the comments "The event was thus surprising in multiple ways. This might happen when we experience a new extreme, but given that Dutch flood risk has safety standards up to once per 100,000 years (Ministry of Infrastructure and Environment, 2016) one would have hoped this to be less of a surprise." and "While most studies aimed at obtaining better estimates of discharge extremes use hydrological or statistical modeling, some follow the approach of using expert judgment (EJ).", please note that this is a must point in every scientific application, since when non-experts apply methods they do not understand, it could lead to failure regardless the magnitude of the selected return period. | As mentioned in the response to comment 2, we now mention that all modelling involves (or at least should involve) expert judgment to some extent. Subsequently we discuss how structured expert judgment with Cooke's method quantifies this process. See lines 55-60. |
|---|---|---|
| 8 | 8) It is mentioned that "For the Dutch rivers Meuse and Rhine, the GRADE instrument is used for this. It generates 50,000 years of rainfall and discharges."; please give more details on this model and how it generates so long rainfall and discharge timeseries (does it use a stochastic simulation approach for the rainfall annual extremes and input these to a hydraulic model to produce the discharge at a specific location in the area of interest?). | The GRADE model is not scientifically published, but it is well described in this report: https://publications.deltares.nl/1209424_004_0018.pdf (Referred to as Hegnauer et al., 2014 in the article). We have added some more details on the GRADE method to the introduction of the article. Please refer to lines 35-44. Note that GRADE is the standard tool in the Netherlands (line 36-37), which is why we use it in this application. |

---

## Author Response (AR2)

Dear Mr. Viviroli and reviewers,

Hereby we submit the second revision of the article: "Using structured expert judgment to estimate extremes: a case study of discharges in the Meuse River". We thank the editor for reconsidering the article, and most of all thank the referees for reviewing the article again.

Compared to the first revision, the changes in this revision are concentrated in the presentation of Cooke's method (referee 2), Bayesian inference (referee 1), and the discussion of the study set-up (referee 2). The major changes thereby are:

- Cooke's method is explicitly presented as a method of estimating uncertainty, including how the experts are evaluated on their performance in estimating uncertainty (rather than the 'true' value. This clarifies Referee 2's comments regarding new/different observations, accidental good estimates, and more or different experts joining the project. Moreover, we refer to literature that researched the added benefits of using performance-based weights over equal weighting.

- The Bayesian approach is described with proper mathematical terminology, and the presentation (Sect. 3.3) is restructured to make the relationship with prior, likelihood, etc. clearer. Furthermore 5.1 discusses the (Renard et al., 2006) article.

The next page contains an overview of the main changes made to the manuscript, ordered by section. A detailed response to both referees' comments is found on the pages thereafter.

Together with this document, we uploaded:

- The new version of the article.

- A comparison between the old and new version using track changes. Note that in addition to the items mentioned above, we revised the full article again and made some minor changes while performing our review. Line numbers or section references are added in the response to the comments to trace where the comments have been processed.

- The (unchanged) supplementary information.

We hope that the manuscript changes further clarify the study to the referees and potential readers alike. Again, we hope that the new version will be reconsidered, and thank the referees again for their effort and input, as their feedback has greatly improved the presentation of our research.

Kind regards,

Guus Rongen
Oswaldo Morales-Nápoles
Matthijs Kok

**Overview of the main changes per section**

1. Introduction
   - Cooke's method (aka the Classical Model) is changed to Classical Model (aka Cooke's method). While they are two different names for the same method, the Classical Model is more consistent with recent scientific literature.
   - Structured expert judgment is introduced without referring to 'everyday' expert judgment.
   - Renard et al., 2006 is added to the examples of a study that uses EJ to limit inform extremes through prior information.
2. Study area and used data.
   - No changes
3. Method description
   - Section 3.1: The three components of the downstream discharge model (Eq. 1)_are listed explicitly, to distinguish the different models used in the study.
   - Section 3.2:
     - the Classical Model is introduced more clearly as a method for estimating uncertainty. The statistical accuracy is explained as (p-value based) method that scores the expert's ability to estimate uncertainty.
     - Literature that compares equal weighting to performance-based weighting out-of-sample is referenced.
   - Section 3.3: This majority of this section was rewritten of restructured to put it in context of Bayes theorem. The method is now addressed with proper Bayesian terminology.
4. Results
   - A comment on how experts are evaluated based on their ability to estimate uncertainty rather than proximity to observed values, in the context of Figure 4.
5. Discussion
   - Section 5.1: A more detailed comparison to Renard et al. 2006 is made, describing why one would choose that method (mainly: eliciting differences) over the method we chose.
   - Section 5.2: A notion is made of GRADE being method that is not published in a peer-reviewed scientific journal.
6. Conclusions
   - Minor edits

| RESPONSE TO REFEREE 2'S COMMENTS (presented in report 1) | |
|---|---|
| **#** | **Referee comment** | **Authors' response** |
| **1A** | 1) I understand the Authors' reply, but I still cannot comprehend what exactly is defined as "an expert's judgment". The traditional approach of an expert's judgment can be based only on models/methods. For example, the Authors mention that "Expert judgment (EJ), in terms of making estimates or verifying observations based on prior knowledge, is often unknowingly applied in everyday practice by researchers and practitioners"; however, in order to make an estimate one requires a model/method and historical observations for fitting/calibration, verification, and validation. Also, what do the Authors mean by "unknowingly"? An expert should know exactly what is (s)he doing and what are the impacts of the applied assumptions. ... | Specifically regarding the sentence: *Expert judgment (EJ), in terms of making estimates or verifying observations based on prior knowledge, is often unknowingly applied in everyday practice by researchers and practitioners,* and *Also, what do the Authors mean by "unknowingly"*:

We have removed this sentence (in the paragraph starting at line 57). It didn't refer to an "expert judgment" as commonly known, but to an everyday estimate or predictions that someone needs to make, which is confusing in this context. |

| | | |
|---|---|---|
| **1B** | …Moreover, I do not entirely agree with this statement and Authors' approach for a different reason; even if the same expert applied different (equally justified) models to the same area (i.e., same case-study, initial/boundary conditions, same input, etc.), then it is certainly expected that the output would be different but not wrong (this is illustrated in the Dimitriadis et al., 2016 study). I think that this procedure is equivalent to the one where multiple (equally qualified) experts used different models/methods in the case study (which is my understanding that this is what the Authors illustrate in their study). However, even if a model/expert is closer to observation does not necessarily mean that this model/expert is better and should be assigned a larger weight coefficient, but (by assuming that all models/experts are equally justified/qualified) that there is an intrinsic uncertainty enclosed in different models/experts, which we should take into account in our flood risk management strategy rather than trying to narrow it down. The danger here is that in a future event, and since all models/experts are equally justified/qualified, the model/expert that was worst in the previous case-study could be now closer to the true observation, and therefore, would be wrong to have assigned a smaller weight coefficient. This would happen because after a limit the uncertainty is intrinsic and can be no longer removed/narrowed but rather only quantified, modelled, and considered in the management strategy. Please note that this is different than applying a wrong model/assumption in a case-study (as explained in my example in the previous review). … | Regarding the rest of the reply, and specifically: *then it is certainly expected that the output would be different but not wrong*", and "*However, even if a model/expert is closer to observation does not necessarily mean that this model/expert is better and should be assigned a larger weight coefficient, but (by assuming that all models/experts are equally justified/qualified) that there is an intrinsic uncertainty enclosed in different models/experts, which we should take into account in our flood risk management strategy rather than trying to narrow it down.*:

First of all, we agree with the referee's comment: …*However, even if a model/expert is closer to observation does not necessarily mean that this model/expert is better and should be assigned a larger weight coefficient, …* We would like to underline that the Classical Model / Cooke's method does not evaluate experts based on their closeness to an observed value, but based on their ability to estimate uncertainty. Fig. 4 in the article illustrates this: expert D and E receive the highest weights because the quantiles/percentiles of their estimates represent the expected fraction (i.e., they are more uniformly distributed). In other words, if we consider an observed value to be randomly drawn from a distribution, and the expert perfectly estimates these distributions, the quantiles shown in Fig 4. will be uniformly distributed (or at least drawn from a uniform distribution). So Cooke's Method evaluates experts based on their ability to estimate uncertainty, and by doing so makes the method relatively insensitive to coincident in the observations. This is addressed with the changes:
- Throughout Section 3.2
- Lines 350 and 351

Regarding the second part: *but (by assuming that all models/experts are equally justified/qualified) that there is an intrinsic uncertainty enclosed in different models/experts, which we should take into account in our flood risk management strategy rather than trying to narrow it down.* A recent study by Cooke et al., 2021 show the added benefit of performance-based weighting in an out-of-sample cross validation (earlier research was based on in-sample cross validation). Lines 202-204 are added to share the main finding of this study (increased informativeness without compromising statistical accuracy), which is the main reason for us to use performance-based weighting over equal weighting.

With respect to the present article, we processed this comment by explained that Cooke's method scores experts based on the product statistical accuracy (SA) and informativeness, where SA is the dominant factor. This is because SA change significantly (orders of magnitude) across experts while informativeness is "more stable".  A statistically accurate expert does not necessarily make estimates close to the seed question's answer (in fact often they don't), but makes estimates that represent the uncertainty in the answers. This is now clarified in two places in Section 3.2. First, with in the explanation of |

|  |  | statistical accuracy, which is explained as being based on a p-value of statistical accuracy, and second: "*The statistical accuracy expresses the ability of an expert to estimate uncertainty. Because a variable of interest is uncertain, its realization is considered to be a value sampled from the uncertainty distribution. According to the expert, this realization corresponds to a quantile on the expert-estimated distribution. If an expert manages to reproduce the ratio of realizations within the interquantile intervals (such as in the example with 20 questions above), the probability of the expert being statistically accurate is high, hence they will receive a high p-value. Of course, this match could* be coincidental, like any *significant p-value from a statistical test. However, in general, a different sample of realizations (in this study, different observed 10-year discharges) is expected to give a p-value (i.e., statistical accuracy) of a similar order.*" (lines 182 to 189) If an expert manages to capture the uncertainty well in their estimates, the statistical accuracy should be similarly high in case the realizations turned out to be different. Just like a statistical test for (e.g.) normality would give a p-value > 0.05 with 95% confidence, if another sample was drawn from a normal distribution. |
|---|---|---|
| **1C** | … If the Authors are certain that all scientists are experts, then I would recommend just quantifying the variability of their judgment/output (i.e., treating them as different 'models', equally correct and justified, as performed in the study by Dimitriadis et al., 2016), and assign an equal weight-coefficient. | In principle, we let the Classical Model decide what the expertise (i.e., uncertainty-estimating ability) of the participants (which we do call experts throughout the study) is, with regard to the questions in this study. However, it is common practice in the Classical Model to present equal weights (EQ) as well. In the main article the EQ results are only shown for the option in which no observations are used in fitting the results. The supplementary material however shows the full results for the EQ decision maker combined with observation (compare Fig 4.2 to Fig 4.3). |

| | | |
|---|---|---|
| 2 | 2) Regarding the reply to the 2nd comment, please present in a clear way what method/model/observations etc., has each expert used to derive his/her results, since "the ability to use one's experience to verify observations." is not a clear definition of an "expert judgment"; for example, what do you mean by "ability"? The only reason I can think of that one expert came up with a different output is that (s)he used different input, initial/boundary conditions, and/or methods in their thinking procedure (as explained in the previous comment). Specifically for the extreme analysis, if, by applying a method/model, the results constantly deviate from observations, then this would mean that the method/model is wrong, should be re-examined, and should be not taken into account in the management risk assessment through the Cookes method (in the recent book by Houstonians, 2022, there are plenty examples how one could severely underestimate the extremes if the assumptions are not correct, as in ignoring dependence, in assigning less robust or even invalid statistical estimators, in applying less accurate statistical distributions, etc.). | For clarity, there are two "models" to be distinguished:
 1. The probabilistic model described in Section 3.1 (Eq. 1) that is used to calculate statistics of downstream discharges using Bayesian inference.
 2. The models that each expert uses to calculate or estimate the components in Eq. 1.
 Model 1 is applied by us (the researchers), based on the experts' estimates from model 2. The performance of model 1 is evaluated in Section 4.4.
 More importantly, what approach the experts used for their estimates (the model 2) was asked during the expert elicitation and is described in *Section 4.2 Rationale for estimating tributary discharges*. We do not know exactly what models the experts have used. However Cooke's Method evaluates the methods used by individual experts (even when we don't know it exactly) based on the statistical accuracy of their answers (which we presume comes from their models). An expert which turned out to give very good estimates as evaluated by the combined score will be assigned a high weight regardless of the methods used (or not) in his/her quantification process. Similarly, if an expert applied a complex hydrological model which results in very bad estimates (for example constantly over or under estimating the seed variables), this indicates that the model is wrong, which results in the expert (and model) not being taken into account by Cooke's Method (as in the reviewer's example).

 To specifically address: *The only reason I can think of that one expert came up with a different output is that (s)he used different input, initial/boundary conditions, and/or methods in their thinking procedure (as explained in the previous comment).*: Differences in estimates would most likely results from differences in their rationale, as the experts were provided with the same information (presented in the supplementary information). Additionally, we know from their description of the applied method (presented in Section 4.3) that their approaches to answer the questions in the study differ. |
| 4 | 4) But what are these components the Authors refer to in their reply and in the manuscript? This is important so that the Readers are able to criticize the experts' methods/models. | The components are now specifically listed at the end of Section 3.1 (lines 149 to 152). As explained in the previous item's response, we do not know exactly what methods were used to make estimates for these components, on top of what is presented in Section 4.3. While we agree that more detail on this would be an interesting addition for the readers, the study focus is not a hydrologic modelling study but on the ability of expert judgments to quantify uncertainty in hydrological problems. In the latter, the performance of their uncertainty estimates is what 'validates' their model. |

| **5** | 5) But what if more experts join this project? More importantly, what if an expert's good judgment (i.e., closer to the true observation) was achieved by accident, and his/her assumptions no longer work for a future event where the conditions have changed? | *But what if more experts join this project?*: We do not know the effect of adding more experts. We do have two experts with a > 0.05 significance level, and with expert D having a SA of 0.683 it is unlikely that additional experts will strongly change the pooled result (but of course we can never know). For context, typically around 5 experts is deemed sufficient (Stephen, 2004 https://doi.org/10.1287/mnsc.1040.0205), just like 10 calibration questions (Colson and Cooke, 2017 https://doi.org/10.1287/mnsc.1040.0205). |

The robustness of the results is typically evaluated in a robustness analysis, in which sensitivity of the statistical accuracy and information score are calculated by leaving one or more experts out at a time and recalculating the measures of interest. The next two tables show the results from this, first for excluding experts, then for excluding items. This shows that the Global Weights DM is overall unsensitive to excluding one expert or item. Note that:

- With respect to experts, the information score and SA are insensitive to specific experts. Note that the SA *increases* when removing the expert with the highest SA (expert D). Expert E (the second expert) becomes dominant, and the small influence of Expert G (third) corrects some of the over- or underestimates, leading to a higher SA.
- With respect to items, the SA is relatively uncertain to specific items, except for excluding the Tabreux 10-year estimate. This is one of the items where Expert D scored significantly better than the other experts. Removing it shifts the weight more to Expert E, which for this particular case results in a lower SA.
- While the tables very similar scores when excluding experts or items, we appreciate that the 1000-year discharge estimates for the decision maker might change significantly if more weight shifts from one to another expert.

| Excluded expert | Information score total | Information score real. | Statistical accuracy |
|---|---|---|---|
| **None** | 0.4852 | 0.4892 | 0.6828 |
| **Exp A** | 0.4338 | 0.4892 | 0.6828 |
| **Exp B** | 0.4852 | 0.4892 | 0.6828 |
| **Exp C** | 0.4389 | 0.418 | 0.6828 |
| **Exp D** | 0.4356 | 0.3913 | 0.7071 |
| **Exp E** | 0.4848 | 0.4892 | 0.6828 |
| **Exp F** | 0.4852 | 0.4892 | 0.6828 |
| **Exp G** | 0.4611 | 0.4761 | 0.6828 |

| Excluded item | Information score total | Information score real. | Statistical accuracy |
|---|---|---|---|
| **None** | 0.4852 | 0.4892 | 0.6828 |
| **ChaudfontaineT10** | 0.4924 | 0.5095 | 0.7059 |
| **ChoozT10** | 0.4841 | 0.4866 | 0.7059 |
| **GendronT10** | 0.4885 | 0.4988 | 0.7059 |
| **MartinriveT10** | 0.4854 | 0.4901 | 0.7059 |

| | | | |
|---|---|---|---|
| **SalzinnesT10** | 0.4833 | 0.4844 | 0.7059 |
| **TabreuxT10** | 0.4904 | 0.5039 | 0.3219 |
| **MembreT10** | 0.4857 | 0.4909 | 0.7059 |
| **StahT10** | 0.4706 | 0.449 | 0.7059 |
| **MeerssenT10** | 0.4821 | 0.481 | 0.5927 |
| **GochT10** | 0.4882 | 0.4979 | 0.4048 |

We did not include this robustness analysis in the main article, as it draws the focus too much to the details of the expert-elicitation. However, if the reviewer would find it suitable, we could include it as an appendix.

*what if an expert's good judgment (i.e., closer to the true observation) was achieved by accident*: Because experts are not assessed by the distance between, for example, the median and the realization but by their statistical accuracy (based on the number of answers to the calibration variables falling in each interquantile interval), the results should be relatively sensitive to a value (such as the median) that is accidentally close to the realization. Note also that the statistical accuracy is based on the total of estimates. For illustration, consider Figure 4. As long as an answer does not change interquantile interval, the SA will not change.

*and his/her assumptions no longer work for a future event where the conditions have changed?* Indeed Cooke's method relies heavily on the assumption that good statistical accuracy for seed variables (10-year return discharge estimates) is also a good statistical accuracy for variables of interest such as 1,000-year return discharges (or for future 10-year discharges, but this should be solved by applying results within the proper context). This is discussed in Section 5.2. "*An implicit assumption is that the experts' ability to estimate the seed variables (a 10-year discharge) reflects their ability to estimate the target variables (a 1000-year discharge). This assumption is in fact one of the most crucial assumptions in Cooke's method and has extensively been discussed in for example Cooke (1991).*" (lines 487 to 490)

| | | |
|---|---|---|
| **8A** | 8) I understand the Authors' reply and I am aware of the GRADE model. However, please understand that it is difficult to trust non-published material, when also, at the same time, hundreds of scientists struggle to find better and more accurate mathematical models to generate long-range rainfall and discharge timeseries. Also, it is clear from the results that the GRADE performed equally (if not better) than the experts' methods/models, and therefore, experts should base their judgment on this model to improve their own judgment. ... | Regarding GRADE being non-published (at least the full method, the weather generator is): We appreciate this point. We added a note to this is the discussion (Section 5.2): "Finally, note that the full GRADE-method is not published in a peer-reviewed journal (the weather generator is, (Leander et al., 2005)). However, because the results are widely used in the Dutch practice of flood risk assessment (and known to the experts as well) we considered them the most suitable source for comparing the results in the present study." (lines 509 to 512) |

| 8B | ... Also, it is clear from the results that the GRADE performed equally (if not better) than the experts' methods/models, and therefore, experts should base their judgment on this model to improve their own judgment. | We did not explicitly handed the individual experts the GRADE statistics (or any other discharge statistics), in order to: 1) avoid influencing their estimates and 2) being able to compare the results to GRADE. As the expert study took place just after a 'new extreme', one may speculate that the experts' results would probably have been "GRADE plus some factor" if they knew the GRADE numbers. The study focuses on evaluating the method, in a case study to the Meuse River. We fully agree that the GRADE results could help improve the experts' estimates in case the study goal was to derive new Meuse EV-statistics. However, for our purpose, we did not present them with these data. |

| RESPONSE TO REFEREE 1'S COMMENTS (presented in report 2) | |
|---|---|
| # | Referee comment | Authors' response |

| # | Referee comment | Authors' response |
|---|---|---|
| 1 | ...However, I am still skeptical about the fact that the ability to guess the magnitude of small floods implies the ability to guess the magnitude of large ones. I know that the experiment cannot be changed now but I am not satisfied by the discussion of the alternative. The Authors motivate their choice in Section 5.1 by saying that the 10-yr flood is a better target than model parameters because it is "observed". I would say that, as a quantile of the model/distribution, it is not observed, it is a model characteristic such as the moments or parameters. I find the "defense" of the choice in Section 5.1 rather weak. I would suggest indicating that an alternative choice could have been taken and, for example, could be tested in future work. The alternative choice (e.g., Renard et al., 2006, doi:10.1007/s00477-006-0047-4) is possible and, I would say, preferable. I strongly suggest that the Authors read Renard et al. (2006), as suggested in my first review, and discuss that alternative method in this paper... | We agree with the reviewer that the 10-yr flood is a quantile (a model characteristic) and not necessarily observed. Discharge is however a quantity that is used by hydrologists in their everyday work, and does not require a transformation from shape or scale parameter to discharge by the expert In principle, it is a measurable quantity ($m^3/s$ for example). The approach presented in Renard et al. 2006 is indeed an alternative choice of which the authors were not aware when designing the study. We explain the main differences between Renard et al. and our study:

1. Renard et al. 2006 combine different models in their Bayesian estimation: Different distributions are combined, using a stationary, step, or sloped model for the location parameter in time. The authors acknowledge the merit of this method and therefore discuss it in more detail in the discussion. For this specific study, we did not adopt the 'multi-model' approach:
• In terms of a varying location parameter in time: this is outside the scope of our study.
• In terms of distributions, we use the GEV because we selected block maxima (and not peaks over threshold). The Gumbel is a particular case of the GEV-distribution, so we are satisfied with using just the GEV distribution and fitting the (possible Gumbel-) tail to the data.

2. When more than one quantile is used, in Renard et al. 2006 the difference between quantiles is used for any quantile after the first, instead of the quantile itself. This should reduce the dependence between the quantiles and therefore the priors as well. While no proof is provided by Coles and Tawn (1996) or Renard et al. (2006) that the difference in quantiles exhibit less dependence than the quantiles themselves this seems a reasonable assumption if the 1000-yr estimate is considered to be the sum of the 10-yr estimate and the estimate for the difference. In our study, two options are considered:
• The combination of observed maxima and the EJ for the 1,000-year discharge. In this case only on EJ-quantile is used, so the approach is the same.
• Using no observed maxima, but both the 10-year and 1,000-year discharges. In this case using the difference between quantiles would make a difference. Eliciting the differences would however come at the cost of the experts not being able to express their beliefs of the 1,000-year discharge directly in their estimate (which we find important, as explained before).

3. The use of a Jacobian to transform the prior, this is discussed at the last item.

Note that the discussion Sect. 5.1 is adjusted to more properly explain the differences between Renard et al., 2006 and the present study. This mainly concerns the theoretical and practical differences between estimating discharges and differences between discharges. |

| 2 | … Regarding the Bayesian method, the Authors have made two major changes, i.e., using a reasonable prior for the GEV shape parameter, and removing the ad-hoc "weighting" procedure used in the first version of the paper. This is good. However, the language used should be corrected. I've never heard of prior likehood or posterior likehood in Bayesian statistics. I don't think the wording exists, please use proper wording (see e.g., https://en.wikipedia.org/wiki/Bayesian_inference#Formal_description_of_Bayesian_inference or any other Bayesian basics reference). … | We have updated the wording to correspond to formal use, as referred to by the reviewer. Additionally, we restructured Sect. 3.3 to put more focus on what information is used to fill the different parts of Bayes Theorem (prior distribution, likelihood), and how MCMC facilitates estimating the posterior distribution. |

| 3 | ... Besides, since the equation on page 19 of the track-change document (no equation and line numbers there) differs from Eq. (6) in the original paper (the $10/N\_i$ is no more in there), how comes that the results do not vary significantly? I would have liked to have an explanation in the reply to the reviewers (not in the new manuscript, of course). ... | The individual GEV-fits do vary because of the change in method, which is most easily observed by comparing the results in Ch. 3 of the supplementary information in the initial manuscript to those in the supp. Information from the last revision (in which case it is Ch. 4). Compare for example the results for expert C, or expert D for Niers, Goch. |
|---|---|---|

The final results from the EJ decision makers do not vary significantly, because these are based on a weighted combination of individual experts. In some cases, the individual results go up, in others they go down. Moreover, the largest differences are for tributaries that do not contribute to the discharge at Borgharen (as the confluences are downstream of that location).

The downstream results at Borgharen (location of interest in this study) are presented in the supplementary material Sect. 3.2 (initial submission) and Sect. 4.2 (last revision). Here we can see the uncertainties have become wider in the new results, but the medians for GL and EQ are largely unchanged.

To illustrate the effects of changing:
1) The old prior to the geophysical prior
2) Removing the factor 10/N

We show the intermediate step of only changing the prior in the following fit for expert D (high weight), and tributary Geul, Meerssen (downstream of Borgharen):

[Figure]

Note the labels on the left, which show which model choices are related to each plot. Left (orange) shows expert judgment only results, right (red) for expert judgment + observations.

Changing the prior had a relatively limited effect (compare first and second row). Removing the factor 10/N (limiting the

| | | observations' weight in the posterior distribution, compare second and third row) had no effect in the EJ only results (since no observations), and a big effect on the combined results. |
|---|---|---|
| **4** | … Also, since the expert judgment is considered as a prior now, as the Authors claim, the equation on page 19 of the track-change document should express it in terms of model parameters and therefore I would have expected a Jacobian in front of g(F^-1(1-p\|theta)) (see e.g. http://mystatisticsblog.blogspot.com/2018/04/jacobian-transformation-and-uniform.html). | The prior should indeed be expressed as a model parameter term (i.e., $\pi(\theta)$) but the experts are not estimating $\theta$ (or part of the vector) directly. As far as we can see,

The effect of using the Jacobian in Renard et al., 2006 is clearest in the stationary exponential model, in which the Jacobian is the (partial) derivative of the quantile function to $\lambda$: $-\log(1-p)$. This corrects for the fact that the quantile function has a different derivative at different non-exceedance probabilities. In our view, such a transformation would be needed when using for example a non-informed (flat) prior, such as in the example in the link provided by the reviewer, such that a uniform estimate for the quantile would result in a uniform $\theta$. For a non-exceedance probability p=0.999 this would be a larger 'factor' than for p=0.9. However, we prefer the prior $\pi(\theta)$ to follow the expert distribution g regardless of the elicited exceedance probability p.

In summary, we did not change the method based on the Renard et al., 2006 article referred to by the reviewer, mainly because 1) the quantiles were elicited and not the difference between quantiles, 2) non-stationarity was out of scope, and 3) we do not think the use of a Jacobian is needed to transform the expert elicited probability density. We do however acknowledge the merits of the approach and have given it a proper discussion in 5.1, such that readers of the article will not be unaware of the approach. |

---

## Author Response (AR3)

Dear Mr. Viviroli and reviewers,

Hereby we submit the fourth revision of the article: "Using structured expert judgment to estimate extremes: a case study of discharges in the Meuse River". We thank the editor for reconsidering the article, and thank the referee 2 for reviewing the article again.

Compared to the last revision, the changes specifically address the comments of referee 2, which mainly concern the use of structured expert judgment in the hydrological context of this study. We now state specifically that we apply the Classical Model for experts judgment, but without evaluating the assumptions in this method, as this has been extensively done in (referenced) literature. We also mention that our method does not replace of supersede a more typical hydrological or hydraulic modelling approach, but that it can be used as an alternative to estimate uncertainties in extremes. We also adopted the referee's suggestion for improving Figure 5.

A detailed response to the comments is found on the next pages.

Together with this document, we uploaded:

- The new version of the article.

- A comparison between the old and new version using track changes. Note that in addition to the items mentioned above, we revised the full article again and made some minor changes while performing our review. Line numbers or section references are added in the response to the comments to trace where the comments have been processed.

- The (unchanged) supplementary information.

We hope that the manuscript changes clarify the Classical Model for structured expert judgment to the referee and potential readers. We thank the referees and editor for their effort and input, and hope that the changes following from their comments have made this work into an appealing article for the hydrological community.

Kind regards,

Guus Rongen
Oswaldo Morales-Nápoles
Matthijs Kok

| RESPONSE TO REFEREE 2'S COMMENTS (presented in report 1) | | |
|---|---|---|
| **#** | **Referee comment** | **Authors' response** |
| **1** | Regarding the authors' reply in the 1st comment of the previous review: If the experts perfectly estimate one of the three (i.e., 5%, 50%, 95%) quantiles of the 10-year discharge for each tributary, In my opinion, the actual values of the percentiles should be shown/compared, and not what is shown in Fig. 4, which is confusing. First of all, for what percentile (5%, 50%, 95%) are the estimates are shown in Fig. 4 entitled "Seed question realizations compared to each expert's estimates". This Figure shows the uncertainty/distribution (of the 5%, 50%, or 95% percentile) as constructed from 70 values (7 experts times 10 estimates per tributary)? If yes, is it correct to construct the distribution (of the 5%, 50%, or 95% percentile) from all the estimates while some seem to be completely off (i.e., with the exception of the expert D and maybe E, the rest experts seem to have estimates with a low probability of occurrence based on the constructed distribution). | Figure 3 is meant to give a visual representation of the concept of "statistical accuracy" or "calibration" in Cooke's sense. In order to clarify this we have included further clarification related to the calibration score. In lines 178-179 we add "(the quantity $2 \cdot N \cdot \sum i=1,...,4\ si\ \log(si/pi)$ is asymptotically $\chi_3^2$)". In lines 182-184 we added *"Figure 4 is presented to visualize the disagreement between si and pi for this study. This figure will be further discussed in subsection 4.1. For now, it is sufficient to note that the agreement between si and pi is highest for expert D"* |
| **2** | Regarding "Because the goal is to elicit uncertainty, experts estimate percentiles rather than a single value. Typically, these are the 5th, 50th, and 95th percentile.", why not have asked them to also estimate the mean and variance, which are very useful (for example, why calculate the 10-year estimate from these 3 quantiles and through a distribution fitting rather than ask the experts to give at least the mean of their estimates)? | Cooke's method for structured expert judgment is based on the elicitation of quantiles from expert's uncertainty distribution. Other methods may be based on the elicitation of other quantities (moments for example). Investigating those alternative methods is out of the scope of our research. We changed the title of section 3.2 (line 157) to better reflect this to *"Assessing uncertainties with the Classical Model for expert judgments"* |
| **3** | 3) For using the Metalog distribution, the authors state that "This distribution is capable of exactly fitting any three percentile estimate.", but many flexible 3-parameter distributions can be fitted by 3 estimates. Similarly for the ratio, where the log-normal distribution is fitted, I think that these distributions should be used in caution (for example in expressions like "as it is unlikely that the 1,000-year discharge is lower than the highest on record"), since they may confuse the readers thinking that these are the actual distributions estimated in this study for the percentiles and discharge ratios, whereas only a few data are used for the fitting and thus, they do not capture other attributes of the distributions (e.g., its tail, etc.; for example, it is shown that streamflows follow a heavy-tail distribution, and thus, the discharge ratio should have a similar tail definitely heavier from the log-normal's one). | We've added extra clarification as follows: *"Notice that for this research, the Metalog distribution represents the uncertainty distribution of each expert over a particular discharge with a given return period. While it is related to the underlying distribution of extreme discharge it does not make any assumption about this underlying distribution other than the ones expressed by experts through their percentile estimates"* In lines 209 – 212 Regarding the log-normal distribution for the ratio (downstream discharge divided by upstream discharge), we added extra clarification as well: *"The ratio itself does not represent streamflow, so there is no need to assume a heavy tailed distribution as would be expected for streamflow (Dimitriadis et al., 2021)"* In lines 291-292. The elements that contribute to these ratio are explained in Section 3.1, lines 133-137. |

| 4 | 4) I am also concerned about the assumption "An implicit assumption is that the experts' ability to estimate the seed variables (a 10-year discharge) reflects their ability to estimate the target variables (a 1000-year discharge).". The 10-year discharge is a not-so-extreme value, while the 1000-year discharge is considered extreme. It has been shown that streamflows follow a heavy-tail distribution (see, if found useful, the largest performed global analysis in Fig. 11 of https://www.mdpi.com/2306-5338/8/2/59, where streamflow is shown to be almost as heavy-tailed as precipitation, which is known to follow Pareto-tail as indicated and extensively discussed in https://www.itia.ntua.gr/en/docinfo/2000/), and so, an expert may have a rainfall-runoff model that is good only in estimating regular discharges rather than extreme ones (or the other way around) that require a separate rainfall-extreme analysis (since the 1000-year rainfall cannot be easily estimated from the observations). I would recommend reflecting on this issue in the Abstract, Conclusions, and maybe even the Title. | *We have modified the text to make it clear that the purpose of the paper is not to reflect on the underlying assumptions of the Classical method but rather to discuss it's potential in improving hydrological studies of extremes. We changed "This assumption is in fact one of the most crucial assumptions in the Classical Model and has extensively been discussed in, for example, Cooke (1991)." To "This assumption is in fact one of the most crucial assumptions in the Classical Model. The objective of this research is not to investigate this assumption. For an example of a recent discussion on the effect of seed variables on the performance of the Classical Model the reader is referred to Eggstaff et al. (2014). The representativeness of the seed variables for calibration variables has extensively been discussed in, for example, Cooke (1991)."* In lines 496-499 |
|---|---|---|
| 5 | 5) Regarding the "However, an informative prior was added to the shape parameter because, with only expert estimates and no data, two discharge estimates are not sufficient for fitting the three parameters of the GEV-distribution. Additionally, the variance in the shape-parameter decreases with increasing number of years (or other block maxima) in a time series. The 30 to 70 annual maxima per tributary in this study are not sufficient to reach convergence.". These are all discussed and analyzed in Koutsoyiannis 2004 (a,b), where it is suggested (Fig. 5-6 in 2004a and Fig. 10-11 in 2004b) that small sizes of records, e.g. 20–50 years hide the distribution's EV2 shape parameter around 0.15 + 0.05 (e.g., in Fig. 13 of 2004b, as estimated from only the largest-lengthed precipitation records above 100 years).

D. Koutsoyiannis, Statistics of extremes and estimation of extreme rainfall, 1, Theoretical investigation, Hydrological Sciences Journal, 49 (4), 575–590, doi:10.1623/hysj.49.4.575.54430, 2004a.

D. Koutsoyiannis, Statistics of extremes and estimation of extreme rainfall, 2, Empirical investigation of long rainfall records, Hydrological Sciences Journal, 49 (4), 591–610, doi:10.1623/hysj.49.4.591.54424, 2004b. | We thank the reviewer for pointing out these references. They have been added to our paper. In lines 269-270 we write *"Similar observations have been presented before for extreme precipitation in Koutsoyiannis (2004a, b)"* |
| 6 | 6) It is mentioned that "When estimates on uncertain extremes is needed, which cannot satisfactorily be derived (exclusively) from a (limited) data-record, the presented approach provides a means of supplementing this information. Structured expert judgment provides an approach of deriving defensible priors, while | We thank the reviewer for his/her kind words regarding respect for our work. Similarly we respect the reviewers work and have tried to reply to his/her comments the best way we can. We appreciate the reviewers observation that different modelers would try to approach the same problem differently. We agree that professionals will try to use the best tools |

the Bayesian framework offers flexibility for incorporating these into probabilistic results by adjusting the likelihood of input or output parameters.". However, when estimates on extremes are needed, one requires the best statistical approaches in the literature (if direct streamflow records are available) or some, equivalently robust, rainfall-runoff models (if only rainfall records are available) that can capture several hydrodynamic aspects of the selected area (as explained in my previous reviews). From either approaches, one can then estimate the uncertainty of the results from these approaches or models. This is not equivalent (and should not be confused) with some experts using (different or even the same) statistical approaches or models in a robust (or maybe incorrect) manner. Additionally, I would follow a more traditional approach, and see which of the expert(s) seem to achieve (in general or for each tributary) better performances in their predictions (which would mean that they have a better understanding of the area and their applied models/methods), and I would follow their suggestions and not the ones from the rest of the experts that they did not perform well.

I respect the authors' work and I would appreciate their reply to this, which is at the core of their paper.

at their disposal be it models, data collected from the field, experiments (when available) or expert judgments.

Our paper provides a well-executed instance of the Classical Model for expert judgments for estimating uncertainty regarding extreme discharges. We show that a well-executed instance of the Classical-Model combined with Bayesian inference can be one of what researchers may regards as the best tools at their disposal. We don not claim it is the only one and we have modified the conclusion to reflect this.

We change "*When estimates on uncertain extremes is needed, which cannot satisfactorily be derived (exclusively) from a (limited) data-record, the presented approach provides a means of supplementing this information. Structured expert judgment provides an approach of deriving defensible priors, while the Bayesian framework offers flexibility for incorporating these into probabilistic results by adjusting the likelihood of input or output parameters*" to "*When estimates on uncertain extremes are needed, which cannot satisfactorily be derived (exclusively) from a (limited) data-record, the presented approach provides a means (not the only mean) of supplementing this information. Structured expert judgment provides an approach of deriving defensible priors, while the Bayesian framework offers flexibility for incorporating these into probabilistic results by adjusting the likelihood of input or output parameters.*" In lines 554 – 558.

We changed "*In our application to the Meuse River, we successfully elicited credible extreme discharges. However, a case studies for different rivers should verify these findings. Considering the credible results and the relatively manageable effort required, the approach presents an attractive alternative for complex hydrological studies where the uncertainty in extremes needs to be constrained.*" To "*Our research does not discourages the use of more traditional approaches such as rainfall-runoff or other hydrodynamic or statistical models. Considering the credible results and the relatively manageable effort required, the approach (when well implemented) can present an attractive alternative to models that approach uncertainty in extremes in a less transparent way.*" In lines 558 – 562.

| 7 | 7) In Figure 5, please indicate the observed/fitted 50th percentile of the 10-year and the 1000-year (through fitting model) discharges to compare with the experts' estimates. | We have adopted this suggestion and added the 10-year and 1,000-year discharges as derived from the data (the 50th percentile) to figure 5. This illustrates the relative proficiency of expert D and E in estimating discharges (even though it is not about the median, but about the full uncertainty estimate). |
|---|---|---|